# Deficit of mitogen-activated protein kinase phosphatase 1 (DUSP1) accelerates progressive hearing loss

Adelaida M Celaya[1,2]*, Isabel Sánchez-Pérez[1,2,3,4,5†], Jose M Bermúdez-Muñoz[1,2†], Lourdes Rodríguez-de la Rosa[1,2,3], Laura Pintado-Berninches[1,3], Rosario Perona[1,2,3], Silvia Murillo-Cuesta[1,2], Isabel Varela-Nieto[1,2]*

[1]Institute for Biomedical Research "Alberto Sols" (IIBM), Spanish National Research Council-Autonomous University of Madrid (CSIC-UAM), Madrid, Spain; [2]Centre for Biomedical Network Research on Rare Diseases (CIBERER), CIBER, ISCIII, Madrid, Spain; [3]Hospital La Paz Institute for Health Research (IdiPAZ), Madrid, Spain; [4]Biochemistry Department, Faculty of Medicine, Autonomous University of Madrid, Madrid, Spain; [5]Biomedicine Unit UCLM-CSIC, Madrid, Spain

**Abstract** Mitogen-activated protein kinases (MAPK) such as p38 and the c-Jun N-terminal kinases (JNKs) are activated during the cellular response to stress signals. Their activity is regulated by the MAPK-phosphatase 1 (DUSP1), a key component of the anti-inflammatory response. Stress kinases are well-described elements of the response to otic injury and the otoprotective potential of JNK inhibitors is being tested in clinical trials. By contrast, there are no studies exploring the role of DUSP1 in hearing and hearing loss. Here we show that *Dusp1* expression is age-regulated in the mouse cochlea. *Dusp1* gene knock-out caused premature progressive hearing loss, as confirmed by auditory evoked responses in *Dusp1*[−/−] mice. Hearing loss correlated with cell death in hair cells, degeneration of spiral neurons and increased macrophage infiltration. *Dusp1*[−/−] mouse cochleae showed imbalanced redox status and dysregulated expression of cytokines. These data suggest that DUSP1 is essential for cochlear homeostasis in the response to stress during ageing.
DOI: https://doi.org/10.7554/eLife.39159.001

*For correspondence:
acelaya@iib.uam.es (AMC);
ivarela@iib.uam.es (IV-N)

†These authors contributed equally to this work

Competing interests: The authors declare that no competing interests exist.

## Introduction

Hearing loss is the most common form of sensory impairment in humans, affecting over 7% of the world's population (466 million people) according to WHO (*WHO, 2018*). This impairing disability is caused by genetic defects, environmental factors or a combination of both. Genetic hearing loss, including non-syndromic and syndromic deafness, is estimated to account for 50% of cases. Certain mutations can also cause susceptibility to noise-induced damage, presbycusis and other complex forms of hearing loss (*Shearer et al., 1993*).

Clinical heterogeneity reflects the rich cellular and molecular complexity of the cochlea. Indeed the development of the inner ear and the maintenance of hearing throughout life requires the precise spatial and temporal expression of a variety of genes, as well as the strict regulation of the interactome (*Hickox et al., 2017*; *Liu et al., 2014*; *Shearer et al., 1993*). The phenotypic and genetic heterogeneity of hearing loss is still under study, and to this end animal studies and particularly mice model systems are fundamental tools (*Bowl et al., 2017*). Their characterization will help us to develop new therapeutic strategies and to understand the molecular relationship between environmental and genetic factors in the establishment of the different types of sensorineural hearing loss.

Age-related hearing loss (ARHL) is a form of progressive hearing loss and a degenerative process. ARHL arises due to impaired function and consequent cellular loss, both in the peripheral inner ear

structures and in central elements of the auditory pathway (*Lee, 2013*). It is thought to be multifactorial and the result of poorly understood genetic susceptibility and environmental factors, such as exposure to excessive noise, ototoxic chemicals, and medical conditions that exacerbate hearing loss in older people (*Joo et al., 2016*).

Studies in humans and animal models, particularly mice, have pointed to oxidative stress and mitochondrial dysfunction as hallmarks of ageing in several tissues (*López-Otín et al., 2013*), including the cochlea (*Müller and Barr-Gillespie, 2015*; *Ren et al., 2013*; *Uchida et al., 2011*).

Mitogen-activated protein kinases (MAPKs) are intracellular protein Ser/Thr kinases that, upon phosphorylation in response to extracellular stimuli, regulate cellular responses such as proliferation, differentiation, motility and survival. The classical MAPKs include ERK1/2/5, which induce tissue growth and survival, and the stress-activated protein kinases JNK1/2/3 and p38 isoforms (MAPK11/12/13/14), which are mediators of several stress stimuli (*Cargnello and Roux, 2011*; *Gupta and Nebreda, 2015*; *Kyriakis and Avruch, 2012*).

JNK and p38 MAPK transcripts and proteins are expressed in the cochlea from late intrauterine development (*Parker et al., 2015*; *Sanchez-Calderon et al., 2010*). They form part of the adult cochlear response to noise insult (*Jamesdaniel et al., 2011*; *Maeda et al., 2013*) and to ageing (*Sha et al., 2009*). In this connection, their chemical inhibition promotes the survival of cochlear cells after damage (*Tabuchi et al., 2010*; *Wang et al., 2007*).

MAP kinase phosphatases (MKP) belong to the large family of dual-specificity phosphatases. These proteins contain a conserved kinase-interaction motif and specifically dephosphorylate the threonine and tyrosine residues of phosphorylated activated MAPKs, thereby controling the duration, magnitude and spatiotemporal profile of their activities (*Owens and Keyse, 2007*).

MKPs can be subdivided into three classes on the basis of their gene structure, sequence similarity and subcellular localization. The first is formed by inducible nuclear phosphatases and includes MKP1 (DUSP1), PAC1 (DUSP2), MKP2 (DUSP4) and HVH3 or DUSP5, which target all classic MAPK with different substrate affinities. The second class includes the cytoplasmic phosphatases MKP3 (DUSP6), MKPX (DUSP7) and MKP4 (DUSP9), which preferentially inactivate ERK1/ERK2. The third class comprises DUSP8, MKP5 (DUSP10) and MKP7 (DUSP16), which can be both nuclear and cytoplasmic and which target stress-activated MAPKs isoforms (*Owens and Keyse, 2007*). Among the first class, DUSP1 is an inducible and primarily nuclear phosphatase that is the principal regulator of the stress MAPKs (*Franklin and Kraft, 1997*).

MKPs are involved in immune cell function, regulating the inflammatory responses both positively and negatively depending on the specific MKP (*Huang and Tan, 2012*). Their dysregulation has also been associated with different types of cancer and pace of cancer progression (*Bermudez et al., 2010*).

Here, we describe the auditory phenotype of the *Dusp1* null mouse. Our results show that: (i) DUSP1 is expressed in the mouse cochlea with a temporal age-regulated pattern; (ii) DUSP1 deficit leads to a premature onset of hearing loss and to an accelerated progression of the hearing-loss phenotype that is caused by the degeneration and death of the sensory epithelium and spiral ganglion neurons in the cochlea; and (iii) dysregulated oxidative balance and exacerbated inflammatory response are among the mechanisms underlying hearing loss in *Dusp1*-deficient mice.

## Results

### *Dusp1* is expressed and age-regulated in the mouse inner ear

The expression of *Dusp1* was analyzed in cochlear samples from embryonic day (E) 15.5 to one-year-old mice (*Figure 1A*). *Dusp1* transcripts were expressed at all the ages studied, with expression levels increasing by 2-fold at the age of cochlear maturation (2 months) and 4-fold at the oldest age studied (12 months) with respect to those at the earliest embryonic age studied. The expression of *Dusp1*, as well as that of other members of the inducible nuclear MAP kinase phosphatases family, *Dusp2*, *Dusp4* and *Dusp5*, was then studied in cochlear samples of 2-month-old wildtype and *Dusp1* null mice. As expected, *Dusp1* was not expressed in the null mouse, whereas *Dusp2* expression was significantly increased in these mice when compared to wildtype mice (*Figure 1B*). These differences were maintained at the ages of 4–5 and 8–9 months, and at 8–9 months, *Dusp5* expression was also significantly reduced in *Dusp1* null mice compared to wildtype mice (*Figure 1—figure supplement*

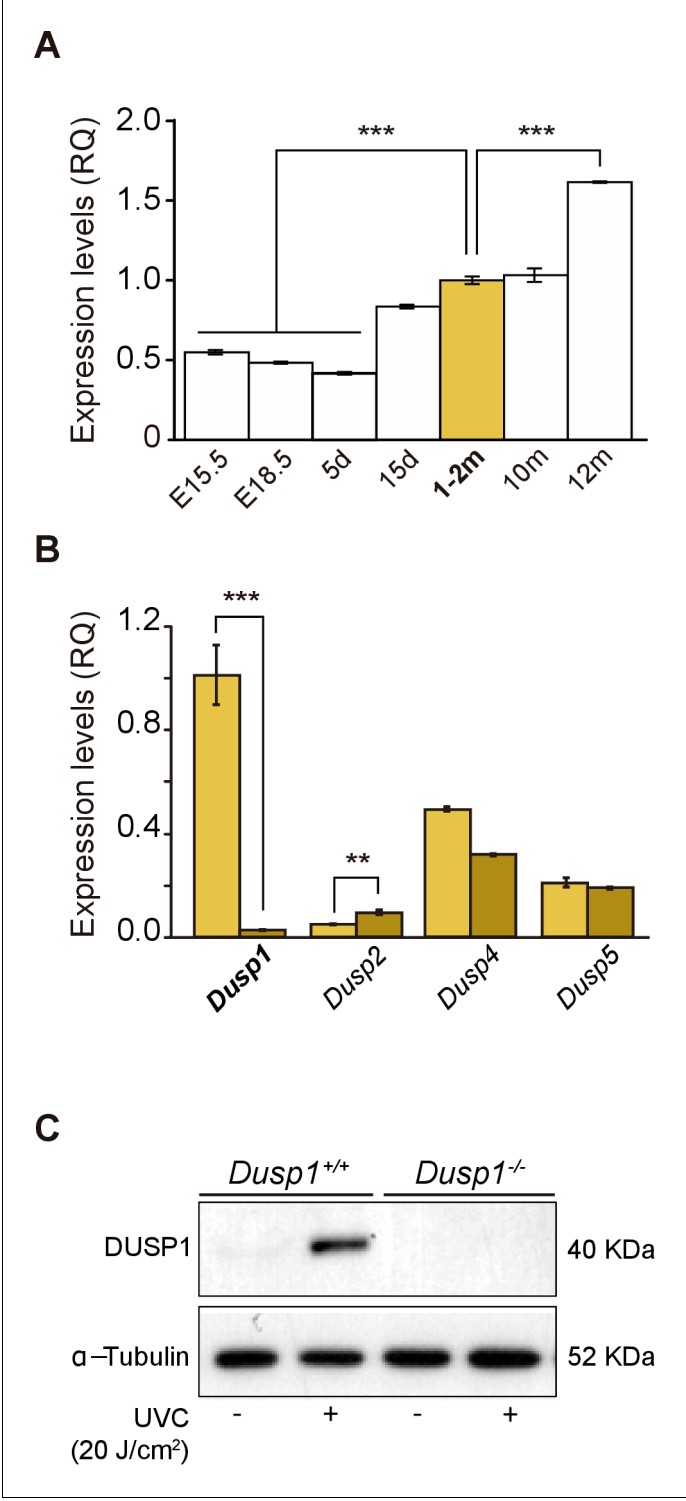

**Figure 1.** Expression of *Dusp1* and related phosphatases in the mouse cochlea. (**A**) Cochlear gene expression of *Dusp1* in MF1 × 129 Sv mice from embryonic (E) to adult stages (measured in days (d) and months (m)). Expression levels were measured by RT-qPCR and calculated as $2^{-\Delta\Delta Ct}$ (RQ), using *Hprt1* as the reference gene and normalized to data from 1–2 month-old mice. Values are presented as mean ± SEM of triplicates from pool samples of three mice per condition. Statistically significant differences were analyzed by Student's t-test, ***p<0.001. (**B**) Cochlear gene expression of inducible nuclear MKPs in the cochlea of 2-month-old *Dusp1*$^{+/+}$ (light yellow) and *Dusp1*$^{-/-}$ mice (dark yellow). Expression levels were calculated as $2^{-\Delta\Delta Ct}$ (RQ), using *Rplp0* as the reference gene and normalized to 2-month-old wildtype *Dusp1* expression. Data are presented as

*Figure 1 continued on next page*

*Figure 1 continued*

mean ± SEM of triplicates from pool samples of three mice per condition. Statistically significant differences were analyzed by the Student's t-test (**p<0.01 and ***p<0.001). (**C**) MEFs cells from *Dusp1*[+/+] or *Dusp1*[−/−] mice were treated or not with 20 J/cm$^2$ UVC light and harvested 30 min after stimuation. 20 µg of whole cell extracs (WCE) were resolved in SDS-PAGE and DUSP1 expression was detected using a specific antibody. Tubulin was used as a loading control.

DOI: https://doi.org/10.7554/eLife.39159.002

The following figure supplement is available for figure 1:

**Figure supplement 1.** Expression of MAP kinase and MAP kinase phosphatases in the mouse cochlea.

DOI: https://doi.org/10.7554/eLife.39159.003

*1A*). Furthermore, we confirmed both the absence of DUSP1 and its induction by stress stimuli in Mouse embryonic fibroblast (MEF) cells. E13.5 MEFs were prepared from wildtype and null mice and exposed to UVC light for 30 min. As expected, MEFs prepared from wildtype mice but not those from null mice showed a rapid induction of DUSP1 levels following stress (*Figure 1C*).

On the other hand, no further differences between genotypes were found in the expression of other MKPs (*Figure 1—figure supplement 1A–C*), nor in the age-evolution of the expression profiles of MAPKs 11 to 14 (*Figure 1—figure supplement 1D*).

## Expression of *Dusp1* is necessary for maintenance of hearing

Hearing was assessed in 1- to 12-month-old mice by evaluating the auditory brainstem response (ABR) in longitudinal experiments (*Supplementary file 1*). Null mice showed elevated ABR thresholds from the age of 2 months when compared to wildtype littermates, although 2-month thresholds were in the range of normal-hearing, showing the typical five waves in response to click sounds (*Figure 2A*). However, the evolution of null mice hearing thresholds worsened with age in response to click and, particularly, to high frequencies presented in tone pure bursts (*Figure 2B*). Thus, we observed premature hearing loss that progressed rapidly from moderate (4–5 months) to profound (8–9 months onward) and affected hearing of initially high and later low frequencies. At the age of 12 months, null mice showed cophosis and therefore no further tests were carried out beyond this age. By contrast, wildtype mice showed cophosis at later ages, between 16–19 months (n = 3, data not shown). Hearing was also assessed in 2-, 5- and 8-month-old *Dusp1* heterozygous mice, no differences in ABR thresholds were found when comparing with wildtype mice (data not shown).

The speed of transmission of the auditory signal was studied by analyzing the latency of appearance of the successive ABR waves elicited by the click stimulus, which was presented at a range of low to high intensities. Null mice showed a significant delay on wave I latency from the age of 2 months (*Figure 2—figure supplement 1*), which was maintained in 4–5- and 8–9-month-old mice (*Figure 2C*). Latencies of waves II and III, but not of IV, also showed significant delays in null mice at the latest ages studied (data not shown). However, null mice showed a significant decrease in the values of I–IV and II–IV interpeak latencies from the age of 4–5 months onwards when compared to wildtype mice of matched ages (*Figure 2D*). No differences were found between genotypes in interpeak latencies in young (2-month-old) mice. Still, even young null mice showed delay on wave I appearance and decreased amplitudes of wave I and IV, which continued for wave IV at 4–5 months of age (*Figure 2—figure supplement 1*).

These data taken together indicate that *Dusp1* null mice show a premature and progressive sensorineural hearing loss.

## Progressive hearing loss of *Dusp1*[−/−] mice correlates with cochlear cellular alterations

Study of the gross anatomy of the middle ear ossicles and inner ear of 2-month-old mice showed no evident morphological alterations between genotypes (*Figure 3—figure supplement 1*). The subsequent morphological evaluation of cochlear sections of mice of both genotypes indicated that 2-month-old mice showed no evident differences, and a normal cytoarchitecture in the basal and middle turns of the cochlea was observed (*Figure 3—figure supplement 2*). Three months later, young adult wildtype mice maintained unaltered cochlear morphology (*Figure 3A,a–d*), whereas *Dusp1* null mice already showed loss of hair and supporting cells in the organ of Corti, loss of neural cells of the

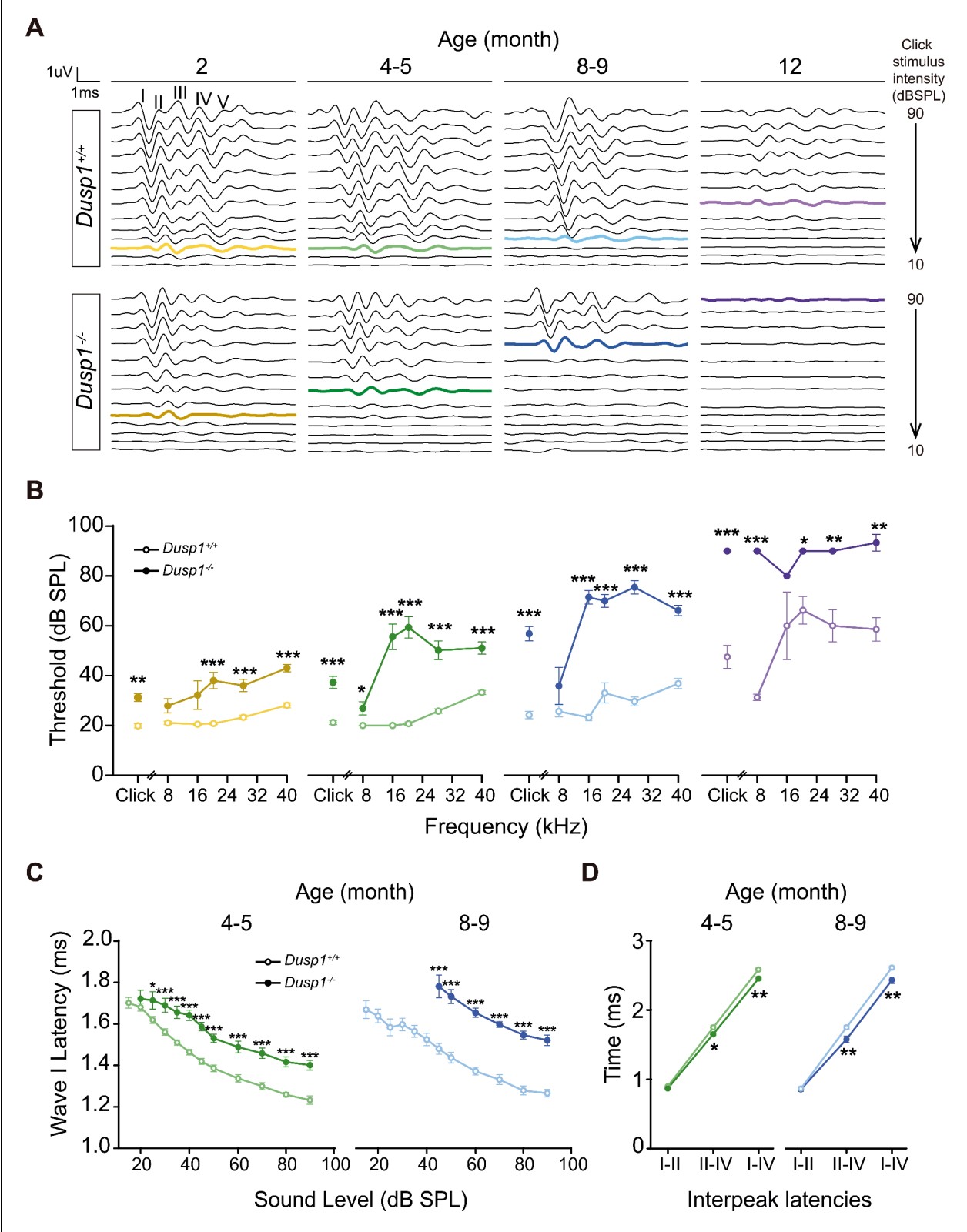

**Figure 2.** Comparative longitudinal hearing evaluation of *Dusp1⁺/⁺* and *Dusp1⁻/⁻* mice. (**A**) Representative ABR recordings showing the typical I–V waves obtained in response to click stimuli from *Dusp1⁺/⁺* and *Dusp1⁻/⁻* mice of 2, 4–5, 8–9 and 12 months of age, showing the hearing thresholds (colored bold lines). (**B**) Evolution of ABR thresholds (mean ± SEM) in response to click and tone burst stimuli in *Dusp1⁺/⁺* (lighter color lines) and *Dusp1⁻/⁻* (darker color lines) mice of 2 (*Dusp1⁺/⁺*, n = 26; *Dusp1⁻/⁻*, n = 25), 4–5 (*Dusp1⁺/⁺*, n = 25; *Dusp1⁻/⁻*, n = 24), 8–9 (*Dusp1⁺/⁺*, n = 24; *Dusp1⁻/⁻*,

*Figure 2 continued on next page*

*Figure 2 continued*

n = 22) and 12 months of age (*Dusp1*$^{+/+}$, n = 7; *Dusp1*$^{-/-}$, n = 3). (C) Input–output function of wave I latency, the mean latency (± SEM) of ABR peak one is plotted against sound intensity (dB SPL) for mice of 4–5 (*Dusp1*$^{+/+}$, n = 25; *Dusp1*$^{-/-}$, n = 24) and of 8–9 months of age (*Dusp1*$^{+/+}$, n = 22; *Dusp1*$^{-/-}$, n = 21). (D) Interpeak latency (mean ± SEM) between peaks I–II, II–IV and I–IV obtained at 80 dB SPL click stimulation in mice of 4–5 (*Dusp1*$^{+/+}$, n = 25; *Dusp1*$^{-/-}$, n = 24) and 8–9 months of age (*Dusp1*$^{+/+}$, n = 21; *Dusp1*$^{-/-}$, n = 21). Statistically significant differences were analyzed by Student's t-test comparing genotypes (*p<0.05, **p<0.01, ***p<0.001).
DOI: https://doi.org/10.7554/eLife.39159.004

The following figure supplement is available for figure 2:

**Figure supplement 1.** ABR latencies and amplitudes of *Dusp1*$^{+/+}$ and *Dusp1*$^{-/-}$ mice.
DOI: https://doi.org/10.7554/eLife.39159.005

basal spiral ganglion (*Figure 3A,e–h*) and loss of of spiral ligament fibrocytes (data not shown). Cochlear cell loss showed a base to apex gradient that progressed with age in concordance with the aforementioned worsening of thresholds from high to low frequencies (*Figure 2*). Therefore, 8-month-old null mice showed loss of cells at the organ of Corti in the middle turn of the cochlea, whereas damage to ganglion cells was evident in the basal turn and extended to the middle turn (*Figure 3A,m–p*). By contrast, 8-month-old wildtype mice still conserved a well-preserved cytoarchitecture (*Figure 3A,i–l*). No differences were observed in the apical turn of the cochlea in 8-month-old mice of different genotypes (data not shown).

Cellular loss was further confirmed by testing the gene expression of molecular markers of specific cochlear cell populations. Thus, *Mpz*, is expressed in the spiral ganglion and cochlear nerve as the major component of the peripheral nervous system myelin sheath (*Wang et al., 2013*). The RNA-binding *RbFox3/NeuN*-coding nuclear factor is expressed in post-mitotic neurons and is involved in the development and maintenance of neuronal functions (*Lin et al., 2016*; *Pan et al., 2017*). The transcription factor *Sox2* is required for inner ear development and is expressed by Schwan cells and supporting cells in the adult cochlea (*Hume et al., 2007*; *Steevens et al., 2017*). Finally, the gene *Prestin* encodes the homonymous motor protein that is specifically expressed in outer hair cells (*Zheng et al., 2000*).

*Figure 3B* shows that *NeuN* transcript levels were significantly decreased in null *Dusp1*$^{-/-}$ compared to wildtype mice over the ages studied. Loss of *Prestin* was also evident, and there was a statistically significant difference between genotypes in 8–9-month-old mice. Furthermore, the TUNEL assay confirmed that cochlear cell loss was apoptotic, thus *Figure 3C* shows apoptotic neural cell death in the basal and middle cochlear turns of 4–5-month-old null mice. At the oldest age studied, 12 months, cell degeneration progressed but showed individual variability, the most damaged null mouse even showed outer hair cell loss in the apical turn (1 out of 3 mice studied), whereas the apical turn in wildtype animals showed no alterations (2 out of 2 mice studied).

A more detailed examination of the organ of Corti of 5-month-old null mice revealed a widespread degeneration of hair cells (myosin VIIa immunoreactivity) and nerve fibers (neurofilaments) that was not observed in wildtype mice (*Figure 4A*). A count of the hair cells on whole-mount preparations confirmed the loss of hair cells in null mice, revealing significantly fewer outer hair cells (OHC) and also inner hair cells (IHC) at the basal turn of the organ of Corti (*Figure 4B*). DPOAE recordings of OHC activity further confirmed cell loss and showed significant threshold increases in null compared to wildtype mice from the age of 4–5 months (*Figure 4C and D*). OHC malfunction progressed with age and maintained a significant difference between genotypes. The study of the distortion product otoacoustic emissions (DPOAE) input–output (I/O) functions of emission amplitudes evoked by f2 = 10.9 and 15.2 kHz is shown in *Figure 4D*. DPOAE I/O functions were significantly lower for 4–5-month-old null mice in the highest frequency studied, and the differences between genotypes progressed to lower frequencies as the mice aged (*Figure 4D*).

## *Dusp1* deficit leads to an early redox imbalance in the cochlea

To further understand the molecular mechanisms underlying the pathological hearing loss phenotype caused by the deficiency in *Dusp1*, we studied oxidative stress and inflammation as two of the main hallmarks of ageing (*López-Otín et al., 2013*) that are closely related with the activity of DUSP1. The oxidative status was evaluated first by measuring the expression levels of genes involved in redox regulation (*Figure 5A*). At the molecular level, by the age of just 2 months,

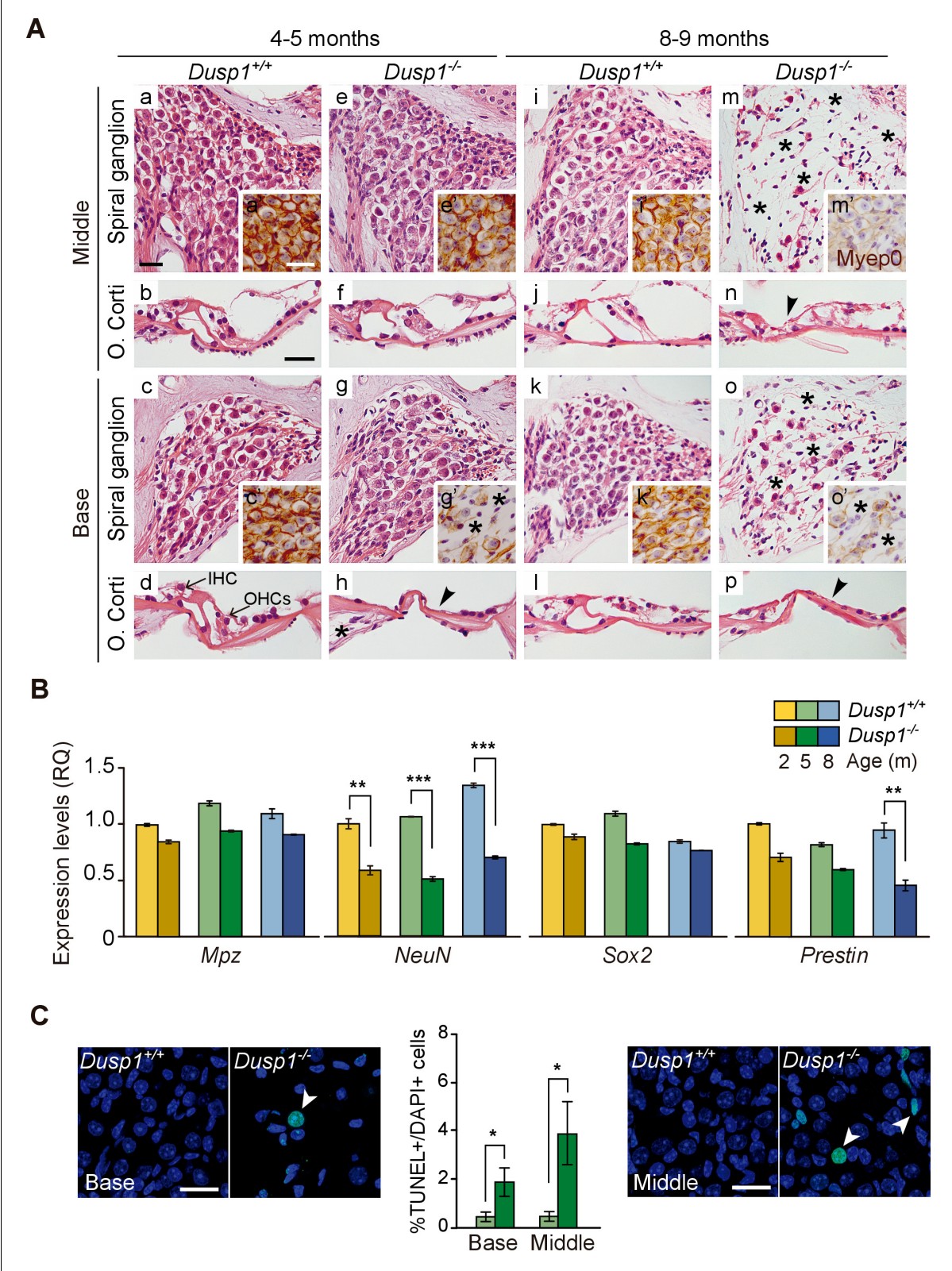

**Figure 3.** Comparative cochlear cytoarchitecture of *Dusp1+/+* and *Dusp1−/−* mice. (**A**) Representative microphotographs of hematoxylin-eosin-stained paraffin cochlear mid-modiolar sections of *Dusp1+/+* and *Dusp1−/−* mice, showing the spiral ganglion and the organ of Corti from the middle and basal turns of the cochlea at 4–5 (n = 5 per genotype) and 8–9 months of age (n = 5 per genotype). Insets present representative microphotographs of myelin protein p0 immunohistochemistry from the middle and basal turns of the cochlea at 4–5 (n = 3 per genotype) and 8–9 months of age (n = 3 per

*Figure 3 continued on next page*

*Figure 3 continued*

genotype). Asterisks and arrowheads indicate the absence of neural and hair cells, respectively. Scale bars: 25 μm. IHC, inner hair cell; OHC, outer hair cell. (B) Cochlear gene expression of *Mpz*, *NeuN*, *Sox2* and *Prestin* in *Dusp1*$^{+/+}$ (lighter color bars) and *Dusp1*$^{-/-}$ mice (darker color bars) at 2, 4–5 and 8–9 months of age. Expression levels were measured by RT-qPCR and calculated as $2^{-\Delta\Delta Ct}$ (RQ), using *Rplp0* as reference the gene and normalized to the 2-month-old wildtype mice group. Values are presented as mean ± SEM of triplicates from pool samples of three mice per condition. Statistically significant differences were analyzed by Student's t-tests comparing genotypes (**$p < 0.01$ and ***$p < 0.001$). (C) TUNEL apoptosis detection. Representative confocal maximal projection microphotographs show the spiral ganglion of the middle and basal turns of 4–5 month-old *Dusp1*$^{+/+}$ (light green bars, n = 4) and *Dusp1*$^{-/-}$ (dark light bars, n = 3) mice. Arrowheads indicate positive TUNEL cells. Quantification of positive TUNEL cells is shown as the percentage of total DAPI-positive cells in a region of interest (ROI) in the spiral ganglion. Values are presented as mean ± SEM. Statistically significant differences were analyzed by Student's t-tests comparing genotypes (*$p < 0.05$). Scale bar: 25 μm.

DOI: https://doi.org/10.7554/eLife.39159.006

The following figure supplements are available for figure 3:

**Figure supplement 1.** Middle and inner ear morphology of *Dusp1*$^{+/+}$ and *Dusp1*$^{-/-}$ mice.

DOI: https://doi.org/10.7554/eLife.39159.007

**Figure supplement 2.** Comparative cochlear cytoarchitecture of 2-month-old *Dusp1*$^{+/+}$ and *Dusp1*$^{-/-}$ mice.

DOI: https://doi.org/10.7554/eLife.39159.008

differences were observed between genotypes in a subset of the genes studied. Transcripts of enzymes involved in glutathione homeostasis(*Gpx1* [1.8-fold] and *Gsr* [1.5-fold]) and in glutathione synthesis(*Gclm* [1.5-fold]) showed a significant increase in null mice. However, no induction of the expression of these genes was found at the other ages studied (*Figure 5A*, first and second rows). Mitochondrial *Ucp1* showed decreased (1.7-fold) and NADPH oxidase components increased (*Nox3* [3-fold] and *Cyba* [1.4-fold]) expression levels in 2-month-old null mice compared with wildtype mice of the same age. *Cyba* transcripts equalized at the age of 5 months between genotypes, but cochlear *Ucp1* and *Nox3* dysregulation was observed at all the ages studied in null mice. No differences between genotypes were observed in *Nox4* expression levels (*Figure 5A*, third row). Finally, analysis of the apoptosis-related genes *Apaf1* and *Kim1* showed no expression differences between genotypes for *Apaf1* but a 2.2-fold increase in the expression of Kim1 in null mice when compared to wildtype mice at 2 and 5 months (*Figure 5A*, fourth row). Accordingly, 5-month-old null mice showed decreased cochlear levels of the mitochondrial antioxidant manganese superoxide dismutase (MnSOD, 1.5-fold). As well as increased levels of P22phox (1.4-fold), of the beta isoform of apoptosis regulator BCL-2-associated X (BAXβ, 1.7-fold) and of phosphorylated p38 (1.4-fold) with respect to wildtype mice (*Figure 5B*). No significant differences were observed in the activation levels of JNK or ERK.

These data suggested that DUSP1 deficiency generates redox imbalance in young mice, which can progressively trigger inflammation and apoptotic cell death. Reinforcing this hypothesis, we observed stronger 3-nitrotyrosine (3-NT) immunoreactivity, a consequence of the nitration of tissue proteins by free radicals, in neural cells of the spiral ganglion in 2-month-old null mice (*Figure 5—figure supplement 1*).

To confirm that the absence of DUSP1 increases the levels of reactive oxygen species (ROS), we next studied the response of MEF cells derived from *Dusp1*$^{+/+}$ and *Dusp1*$^{-/-}$ mice to the oxidant stimulus $H_2O_2$. Our data showed that $H_2O_2$ strongly increased ROS levels in *Dusp1*$^{-/-}$ compared to *Dusp1*$^{+/+}$ cells (*Figure 5—figure supplement 2A*). These results reinforced the theory that the absence of DUSP1 modulates oxidative stress in MEF cells. To further assess the possibility that DUSP1 deficiency leads to DNA damage and, eventually, to cell death, we next quantified γ-H2AX-associated foci in MEF cells from both genotypes. The results showed that cells derived from *Dusp1*$^{-/-}$ mice showed more foci per cell than *Dusp1*$^{+/+}$ cells, indicating DNA damage in basal conditions (*Figure 5—figure supplement 2B*).

## *Dusp1* deficit triggers an exacerbated inflammatory response

Subsequently, we studied the cochlear expression of genes that encode proinflammatory and anti-inflammatory mediators (*Figure 6A*). We found a significant increase in the expression of the anti-inflammatory interleukin 10 gene (*Il10*, 2-fold) together with a decrease in the expression of *Foxp3* (1.4-fold) in 2-month-old null mice. Interestingly, progression of hearing loss is correlated to inflammatory dysregulation, with pro-inflammatory cytokines being strongly upregulated in null mice from

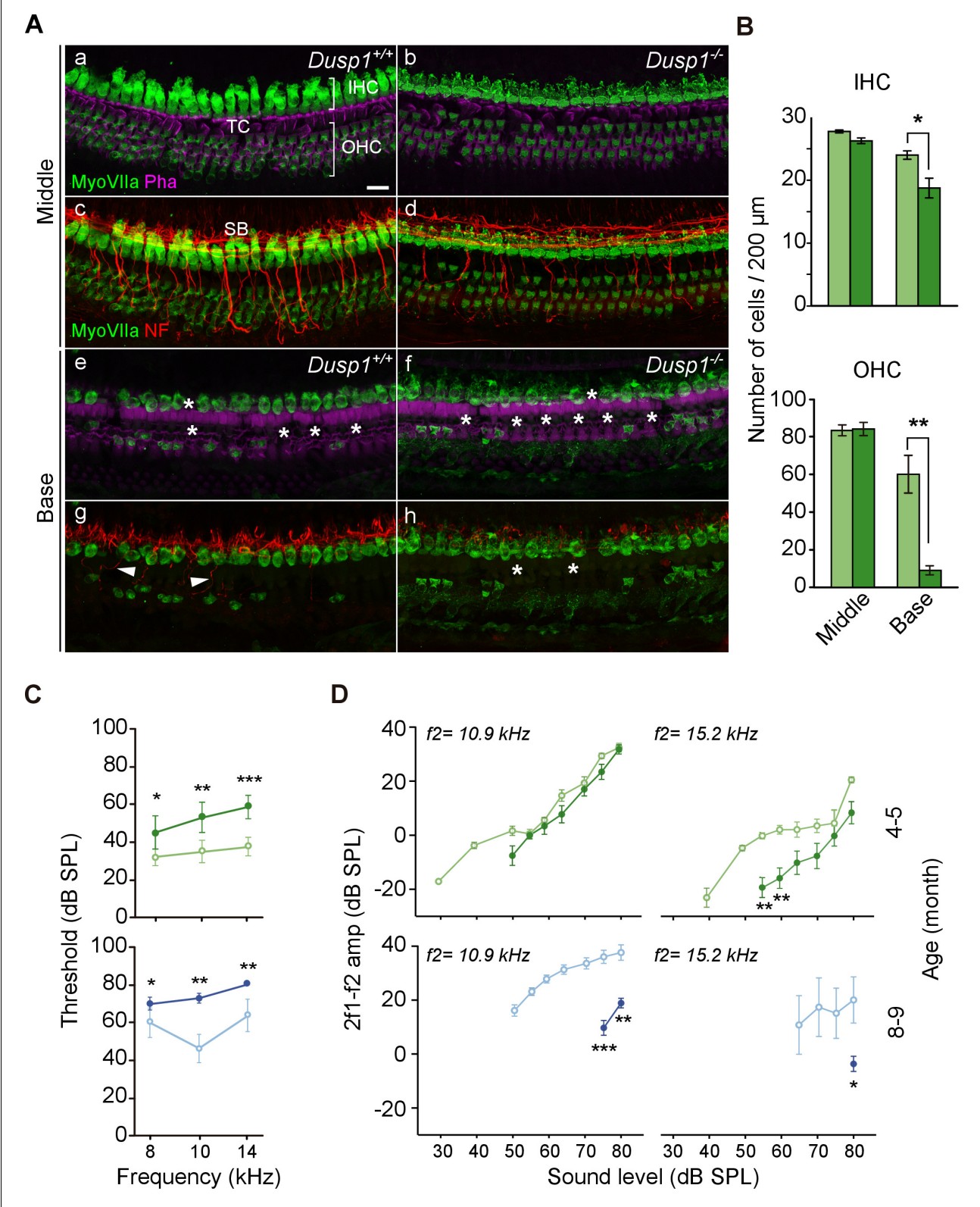

**Figure 4.** Organ of Corti degeneration in *Dusp1⁻/⁻* mice. (**A**) Representative confocal maximal projection images of the organ of Corti of the middle and basal turns of 5-month-old *Dusp1⁺/⁺* (n = 4) and *Dusp1⁻/⁻* (n = 4) mice immunolabeled for hair cells MyoVIIA (green), neurofilament (red) and phalloidin (purple). Asterisks and arrowheads indicate the absence or presence, respectively, of hair cells and fibers. Scale bar: 10 μm. IHC, inner hair cell; OHC, outer hair cell; SB, spiral bundle; TC, tunnel of Corti. (**B**) Quantification in the middle (35–45% distance from apex) and basal cochlear turns

*Figure 4 continued on next page*

*Figure 4 continued*

(60–70% distance from apex) of 5-month-old $Dusp1^{+/+}$ (light green bars) and $Dusp1^{-/-}$ (dark green bars) mice of the number of outer (base, n = 3 per genotype; middle, n = 4 per genotype) and inner hair cells (base, n = 4 per genotype; middle n = 4, per genotype). Values are presented as mean ± SEM. (C) DPOAE thresholds (mean ± SEM) of $Dusp1^{+/+}$ (light color lines) and $Dusp^{-/-}$ (dark color lines) mice of 4–5 months of age (8 kHz: $Dusp1^{+/+}$, n = 5, $Dusp1^{-/-}$, n = 8; 10 kHz: $Dusp1^{+/+}$ n = 4, $Dusp1^{-/-}$, n = 8; 14 kHz: $Dusp1^{+/+}$, n = 4, $Dusp1^{-/-}$ n = 7) and 8–9 months of age (8 kHz: $Dusp1^{+/+}$n = 5, $Dusp1^{-/-}$, n = 5; 10 kHz: $Dusp1^{+/+}$ n = 4, $Dusp1^{-/-}$ n = 5; 14 kHz: $Dusp1^{+/+}$ n = 4, $Dusp1^{-/-}$ n = 4). (D) DPOAE amplitude I/O function (mean ± SEM) evoked by stimulus (f2 = 10.9 kHz or f2 = 15.2 kHz) of $Dusp1^{+/+}$ (lighter color lines) and $Dusp1^{-/-}$ (darker color lines) for mice of 4–5 and 8–9 months of age (at least three mice per genotype). Statistically significant differences were analyzed by Student's t-tests comparing genotypes (*p<0.05, **p<0.01, ***p<0.001).

DOI: https://doi.org/10.7554/eLife.39159.009

the age of 5 months. Thus *Il1b*, *Tnfa* and *Tgfb1* expression levels were increased by 1.7, 1.6 and 1.6-fold, respectively, in the null mouse. By contrast, *Il6* showed no expression differences and increased with ageing similarly in both genotypes.

Infiltration by IBA1[+] macrophages is part of the inflammatory and phagocytic cochlear response to damage. Five- and 8-month-old null mice showed more macrophage infiltration than did wildtype mice (*Figure 6B*, compare the first two columns), with IBA1[+] cells in the spiral ligament in a gradient from base to apex (*Figure 6B*, quantification in Figure6C), and a progression with age (*Figure 6B*, compare central and right columns).

To further study the contribution of DUSP1 in inflammation, MEFs prepared from wildtype and null mice were treated with TNFα (10 ng/ml) for different periods of time (*Figure 6—figure supplement 1*). The results showed that TNFα induced p38 in both genotypes, but that the molecular apoptotic marker Caspase 3 is cleaved (activated) only in $Dusp1^{-/-}$ mice 4 hr after treatment (*Figure 6—figure supplement 1*).

## DUSP1 deficit increases and MAPK14 deficit reduces noise-induced hearing loss

Exposure to noise accelerates the loss of hearing that is associated with ageing. ARHL and noise-induced hearing loss (NIHL) are associated with increased ROS production, inflammation and OHC apoptosis (*Kurabi et al., 2017*; *Wong and Ryan, 2015*). To test whether DUSP1 deficit influenced the extent of noise-induced insult, 2-month-old *Dusp1* null and wildtype mice were exposed to noise. Three days later, the *Dusp1* null mice were severely damaged, especially in frequencies over 8 kHz, when compared with wildtype mice (*Figure 7A*). Differences between genotypes at 16 kHz and 20 kHz were maintained 14 days after noise exposure. $Dusp1^{+/-}$ heterozygotes showed a response similar to that of wildtype mice (data not shown). To confirm that noise induces DUSP1 in wildtype mice, cochlear samples were taken 45 and 90 min after noise exposure, and DUSP1 levels were measured by western blotting. Indeed, noise transitorily induced DUSP1 (for around 45 min after the noise). P-p38 was also tested in parallel but, as described, it was not increased at these early post-noise timepoints (*Figure 7B*). Thus, to further confirm that the level of phosphorylation of stress kinases is essential for the progression of noise insult, tamoxifen (TAM)-treated *Mapk14* conditional knockin (KI/KI and +/KI) and wildtype (+/+) mice were also exposed to noise. No differences in hearing thresholds were detected among these genotypes following TAM treatment (*Figure 7C*, left panel, baseline). Mice with a total or partial deficiency in *Mapk14* showed ABR threshold shifts that were smaller than those shown by wildtype mice from the first time tested, suggesting that *Mapk14* activation plays a central role in the progression of noise-induced injury.

## Discussion

In this work, we show that *Dusp1* plays a key role in hearing maintenance and, therefore, its deficit accelerates progressive hearing loss as mice age. DUSP1-deficient mice show progressive hearing loss with age, and cochlear cellular degeneration progresses with age from the cochlear base to the apex.

DUSP1 has been classified as an inducible nuclear phosphatase but it has also been found in cytosol, mitochondria and peroxisomes (*Kataya et al., 2015*). This protein dephosphorylates MAPK proteins, with substrate specificity for p38 and JNK (*Patterson et al., 2009*). DUSP1 is an early-

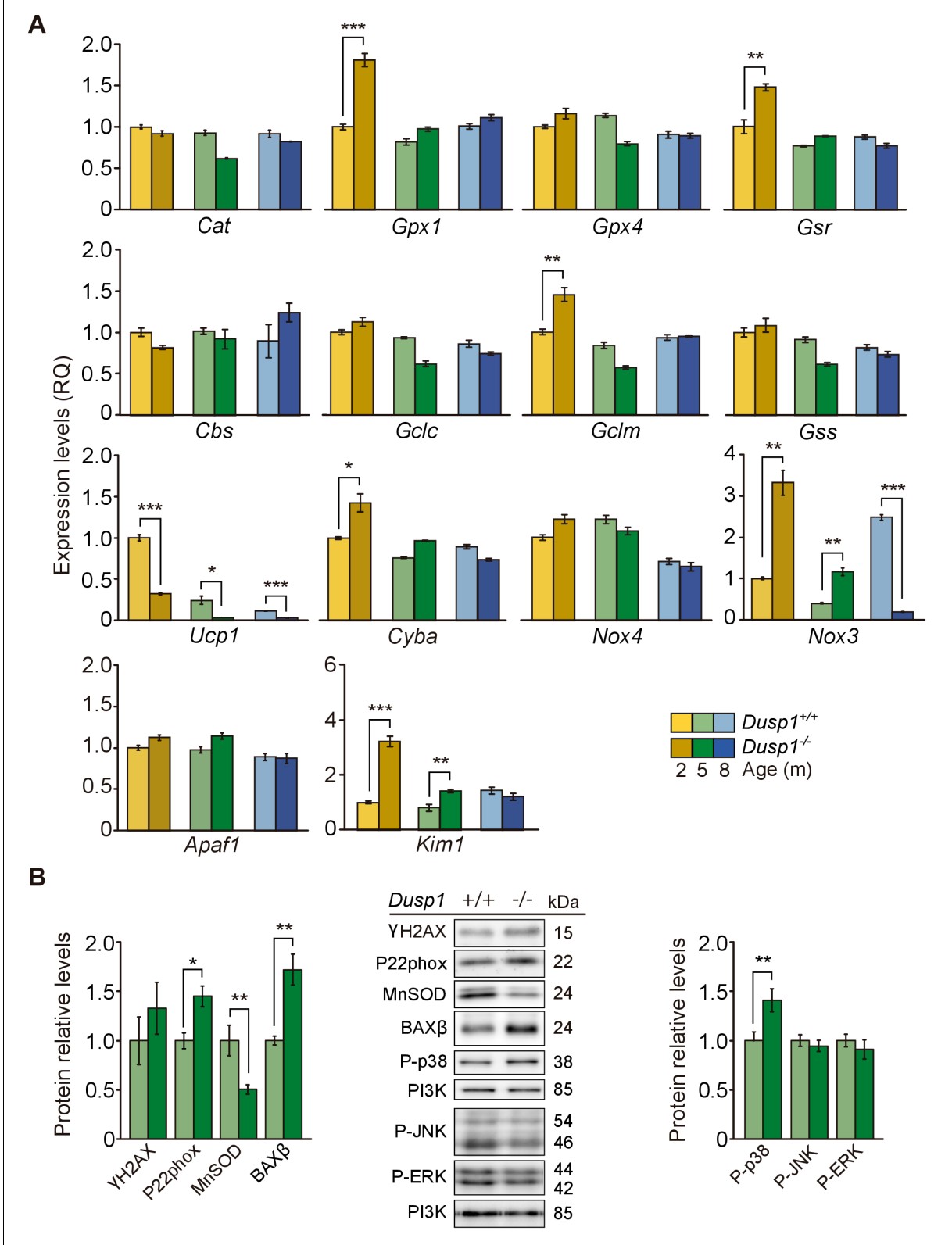

**Figure 5.** Cochlear oxidative stress status of $Dusp1^{+/+}$ and $Dusp1^{-/-}$ mice. (**A**) Cochlear expression of redox regulation and apoptosis genes in $Dusp1^{+/+}$ (lighter color bars) and $Dusp1^{-/-}$ mice (darker color bars) of 2, 4–5 and 8–9 months of age. Expression levels were measured by RT-qPCR and calculated as $2^{-\Delta\Delta Ct}$ (RQ), using $Rplp0$ as the reference gene and normalized to data from the 2-month-old wildtype mice group. Values are presented as mean ± SEM of triplicates from pool samples of three mice per condition. (**B**) Cochlear protein relative levels were measured by western blotting. *Figure 5 continued on next page*

*Figure 5 continued*

Representative blots and quantification of levels are shown for γ-H2AX, P22phox, MnSOD, BAXβ, P-p38, P-JNK and P-ERK1/2 cochlear protein extracts from 5-month-old *Dusp1*$^{+/+}$ (light green bars) and *Dusp1*$^{-/-}$ mice (dark green bars). Expression levels were calculated as a ratio using PI3K as housekeeping protein and normalized to the wildtype mice group. Values are presented as mean ± SEM of triplicates from pool samples of three mice per condition. Statistically significant differences were analyzed by Student's t-tests comparing genotypes (*p<0.05, **p<0.01, ***p<0.001).
DOI: https://doi.org/10.7554/eLife.39159.010
The following figure supplements are available for figure 5:

**Figure supplement 1.** 3-Nitrotyrosine ( 3-NT) immunohistochemistry.
DOI: https://doi.org/10.7554/eLife.39159.011
**Figure supplement 2.** Production of reactive oxygen species and DNA damage in MEF cells.
DOI: https://doi.org/10.7554/eLife.39159.012

response gene that is induced transitorily by different stress-related stimuli (*Gass et al., 1996*). DUSP1 expression is low during development in almost all tissues except liver and intestine. RNAseq studies indicate that it is expressed in the cochlea and vestibule, particularly in supporting cells (data available from gEAR, umgear.org) (*Burns et al., 2015*; *Elkon et al., 2015*; *Liu et al., 2014*; *Scheffer et al., 2015*), and that it is upregulated in the cochlea after exposure to noise (*Alagramam et al., 2014*). In general, MKP family members have been reported to present low basal levels of expression in most tissues (*Camps et al., 2000*) but they are rapidly induced after stress stimuli. Accordingly, we have observed here that DUSP1 expression is induced in *Dusp1*$^{+/+}$ MEF by 30 min of UV irradiation and in *Dusp1*$^{+/+}$ cochlea by 30 min of exposure to noise.

The MKP family includes many members that have broad substrate specificity. Studies using MKP-deficient in vivo models have shown that the deletion of one family member typically results in a complex phenotype, which reflects the complex transcriptional regulation of these proteins by MAP kinases and the compensatory effects of other MKPs (*Huang and Tan, 2012*). In this work, we have observed that *Dusp1* transcripts show their highest expression levels in the cochlea among the nuclear-inducible MKPs, and interestingly that these expression levels increase with age. These data suggest that an increase in DUSP1 expression level is required to maintain hearing with age. Other family members do not seem to compensate for the specific action of DUSP1 in the cochlea, although they could potentially contribute to ameliorating the hearing phenotype through the increased expression of *Dusp2*. DUSP2 is a nuclear phosphatase, enriched in hematopoietic cells, that is specific for ERK and p38. It is a positive regulator of inflammatory response, and so the macrophages of *Dusp2* null mice produce lower levels of pro-inflammatory cytokines. This deficit is accompanied by the decreased and increased activation of ERK and JNK, respectively (*Owens and Keyse, 2007*). However, as none of the MKPs studied here showed expression changes as the mice aged, it seems that there are no alternative compensatory mechanisms to DUSP1 protection in age-related hearing loss.

MKP family members *Dusp6*, *Dusp7* and *Dusp9*, which mainly target ERK1/2, have been reported to be expressed during inner ear development (*Urness et al., 2008*). Indeed, the inactivation of *Dusp6* leads to hypoacusis that results from malformations in the middle ear and otic capsule but not in the cochlea (*Li et al., 2007*). By contrast, we show here that DUSP1 deficit does not affect either middle ear or otic capsule formation. Indeed, the hearing-loss phenotype of the null mice has a clear correlation with the progressive loss of cochlear cell populations, particularly the loss of the hair and supporting cells of the organ of Corti and of the neural cells of the spiral ganglion. Reported human ARHL cochlear alterations include changes to structural elements such as hair cells, neurons, lateral wall tissues, or a combination thereof (*Kusunoki et al., 2004*; *Nelson and Hinojosa, 2006*). Thus, the cochlear phenotype of the *Dusp1* null mouse reproduces that reported for human ARHL, and shares characteristics with other animal models of progressive hearing loss (*Dubno et al., 2013*; *Espino Guarch et al., 2018*; *Riquelme et al., 2010*), including increased apoptotic cell death in the ageing cochlea (*Sha et al., 2009*).

Our data suggest that the activity of DUSP1 is an essential piece of the molecular mechanism involved in otoprotection during ageing. Indeed, DUSP1 has already been described as neuroprotective (*Collins et al., 2013*; *Taylor et al., 2013*) and as a regulator of neuroinflammatory processes (*Collins et al., 2015*). Low levels of DUSP1 have been associated with neurological pathologies such as Huntington disease, multiple sclerosis, ischemia or cerebral hypoxia (*Collins et al., 2015*).

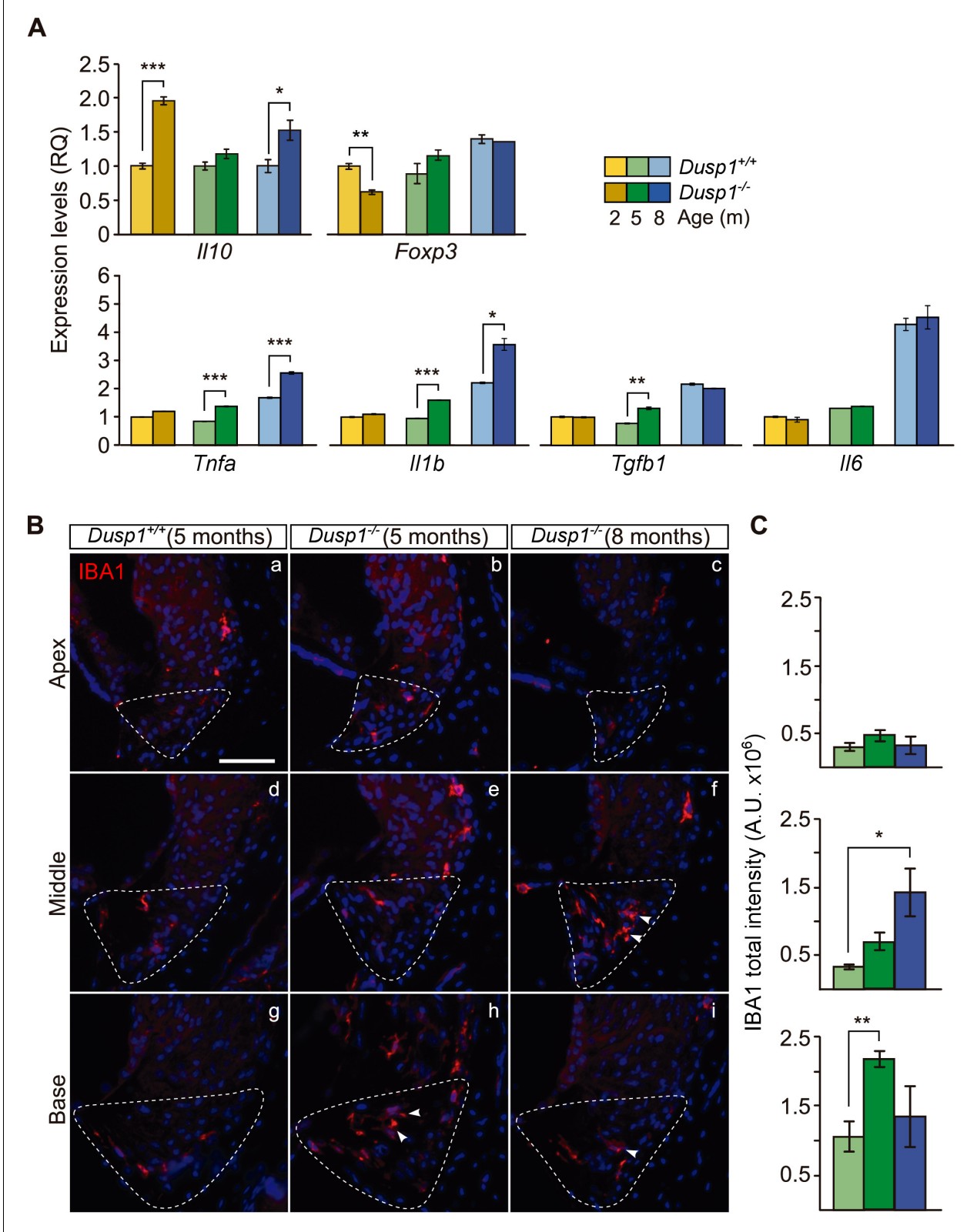

**Figure 6.** Exacerbated inflammatory response in *Dusp1⁻/⁻* mice. (**A**) Cochlear expression of inflammatory response genes in *Dusp1⁺/⁺* (lighter color bars) and *Dusp1⁻/⁻* mice (darker color bars) of 2, 4–5 and 8–9 months of age. Expression levels were calculated as $2^{-\Delta\Delta Ct}$ (RQ), using *Rplp0* as the reference gene and normalized to the 2-month-old wildtype mice group. *Il10* data were normalized to the matched-age wildtype mice groups. Values are presented as mean ± SEM of triplicates from pool samples of three mice per condition. (**B**) Representative microphotographs of cochlear

*Figure 6 continued on next page*

*Figure 6 continued*

mid-modiolar cryosections immunolabeled for IBA1, showing a detail of the spiral ligament of the apical, middle and basal turns of 5-month-old *Dusp1+/+* (n = 3) and *Dusp1−/−* mice of 5 (n = 3) and 8 months of age (n = 3). The Type IV fibrocytes region is outlined. Arrowheads point to macrophage cells. Scale: 50 μm. (C) IBA1 total fluorescence intensity was measured in the spiral ligament in each cochlear turn of 5-month-old *Dusp1+/+* (light green bars; base, n = 4; middle, n = 5; apical, n = 4) and *Dusp1−/−* mice of 5 months (dark green bars, base, n = 4; middle, n = 6; apical, n = 5) and 8 months of age (dark blue bars, base, n = 4; middle, n = 3; apical, n = 3). Values are presented as mean ± SEM. Statistically significant differences were detected by Student's t-test comparing genotypes (*p<0.05, **p<0.01, ***p<0.001).

DOI: https://doi.org/10.7554/eLife.39159.013

The following figure supplement is available for figure 6:

**Figure supplement 1.** TNF-α induces caspase 3 activation in *Dusp1−/−* MEF cells.

DOI: https://doi.org/10.7554/eLife.39159.014

Interestingly, BDNF-induced axon branching is regulated by DUSP1 during development (*Jeanneteau et al., 2010*; *Mullen et al., 2012*). Although 2-month-old null mice already have worse hearing thresholds than their wildtype littermates, we did not observe a gross cochlear developmental defect. Nevertheless, these young mice already show defective temporal processing and responsiveness of auditory signal. As in other murine models of ARHL, at later ages this defect is not compensated for in the central stations of the auditory pathway (*Möhrle et al., 2016*). These differences were not observed in one-month-old null mice, suggesting that young null mice have subtle hearing deficiencies that are most possibly related to alterations in basic molecular mechanisms.

The increased production of ROS found in $H_2O_2$-stimulated MEF from embryonic null mice supports this concept, and points to chronic oxidative stress as one of the possible causes of later problems. This redox imbalance was further confirmed in the cochlea of young null mice by the detection of 3-NT, a metabolite of peroxynitrite action that is increased in the ageing ear (*Jiang et al., 2007*). Indeed, abnormally increased ROS levels play a role in ageing and in the onset and progression of age-related diseases (*Balaban et al., 2005*), including presbycusis (*Yang et al., 2015*). Mitochondria are the source of 90% of intracellular ROS, which are a byproduct of mitochondrial respiration, and one of the main targets of the oxidative damage caused by them. ROS can also be generated by non-mitochondrial sources such as the NADPH oxidases (*Di Meo et al., 2016*). DUSP1 upregulation is induced by ROS, specifically by those produced by NADPH oxidases (*Fürst et al., 2005*), and the transitory expression of this protein can also be stabilized by RNA-binding proteins in contexts of oxidative stress (*Kuwano et al., 2008*). Interestingly, the observed cochlear cell loss follows a gradient from base to apex. It is well known that basal structures are more susceptible in general to insult (*Murillo-Cuesta et al., 2010*; *Sanz et al., 2015*; *Takada et al., 2014*) and in particular to ageing (*Makary et al., 2011*; *Nadol, 1979*; *Ohlemiller et al., 2010*). It has been proposed that higher basal susceptibility to damage is the result of increased ROS generation (*Choung et al., 2009*) and that basal hair cells are highly damaged by free oxygen radicals (*Sha et al., 2001*).

*Dusp1* null mice present an early decrease of mitochondrial uncoupling protein 1 (UCP1), as well as an imbalance of NAPDH oxidases. Uncoupling proteins facilitate the dissipation of the proton gradient needed for oxidative phosphorylation, leading to heat generation and reduced ROS generation by mitochondria (*Kazak et al., 2017*). UCP1–UCP5 are expressed in the spiral and vestibular ganglia and are upregulated by some stress stimuli (*Kitahara et al., 2005*; *Rodríguez-de la Rosa et al., 2015*). On the other hand, the NADPH oxidase NOX3 constitutively generates superoxide, which is further converted to $H_2O_2$, in a p22PHOX-dependent manner. NOX3 is expressed in the vestibular and cochlear sensory epithelium and spiral ganglion (*Bánfi et al., 2004*). NOX3 is downregulated by noise (*Vlajkovic et al., 2013*) and participates in cisplatin-induced superoxide formation and ototoxicity (*Kaur et al., 2016*). Protection from ROS-induced damage is achieved by a complex system of antioxidant enzymes: superoxide dismutase, catalase, glutathione peroxidase, glutathione reductase and peroxiredoxin (*Poirrier et al., 2010*). It has been proposed that, with ageing, the imbalance between increased production and impaired detoxification systems leads to a pro-oxidative state (*Nóbrega-Pereira et al., 2016*). Thus, natural ageing could lead to oxidative damage to hair cells and ganglion neurons, which require great amounts of energy to process sound, ultimately leading to cell and functional loss (*Miller, 2015*).

In fact, oxidative stress and mitochondrial dysfunction are the underlying mechanisms pointed to by many of the uncovered genetic factors associated with ARHL. For example, genetically modified

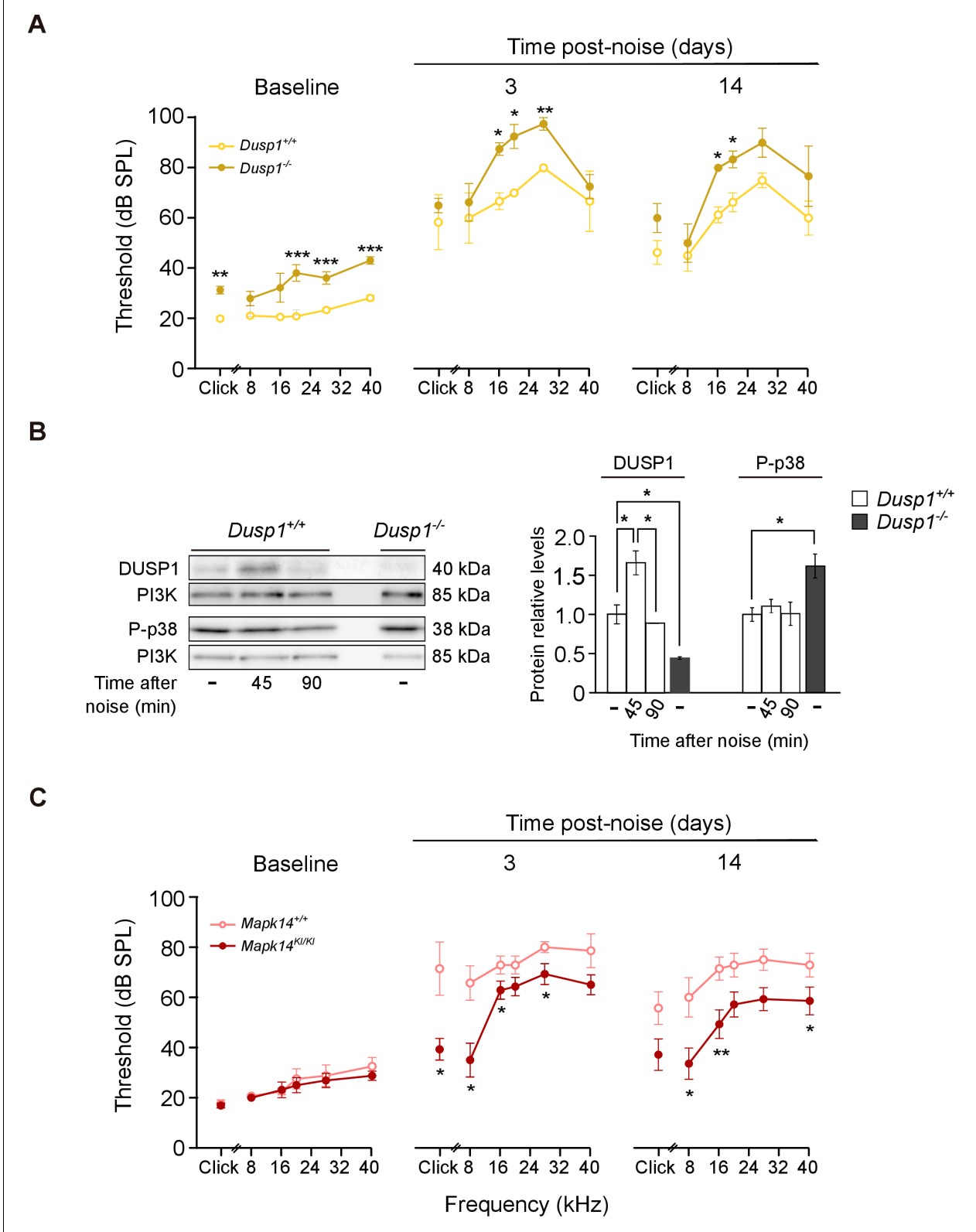

**Figure 7.** Hearing loss following noise-exposure of *Dusp1* and *Mkp14* null mice. (**A**) ABR thresholds (mean ± SEM) in response to click and tone burst stimuli before noise exposure (baseline) and 3 and 14 days post-noise (VSS, 107 dB, 30 min) exposure in *Dusp1*$^{+/+}$ and *Dusp1*$^{-/-}$ mice (n = 4 per genotype). (**B**) Representative blots and quantification of levels are shown for DUSP1 and P-p38 cochlear protein extracts from *Dusp1*$^{+/+}$ and *Dusp1*$^{-/-}$ mice and wildtype mice, 45 and 90 min after noise exposure. Expression levels were calculated as a ratio using PI3K as the loading control. Values are

*Figure 7 continued on next page*

*Figure 7 continued*

presented as mean ± SEM of triplicates from pool samples of three mice per condition. (C) Evolution of ABR thresholds (mean ± SEM) in response to click and tone burst stimuli before (baseline) and 3 and 14 days post-noise exposure in $Mkp14^{+/+}$ and $Mpk14^{KI/KI}$ mice (n = 7 per genotype). Statistically significant differences were analyzed by Kruskal–Wallis tests comparing genotypes (*p<0.05, **p<0.01, ***p<0.001).
DOI: https://doi.org/10.7554/eLife.39159.015

mice that have defects affecting *Gpx1*, *Sod1*, *Polg1*, *mt-Tr*, and *Cs* show traits of ARHL (*Bowl and Dawson, 2015*; *Müller and Barr-Gillespie, 2015*). Human genetic studies of ARHL have also identified polymorphisms of genes in the same functional category, such as *CAT*, *MTHFR*, *MTR*, and *UCP2*, within a single cohort (*Dawes and Payton, 2016*). The glutathione-transferase genes (*GSTM1* and *GSTT1*) (*Bared et al., 2010*; *Van Eyken et al., 2007*) and N-Acetyltransferase 2 (*NAT2*) have also been studied extensively and associated with human ARHL (*Unal et al., 2005*; *Van Eyken et al., 2007*). *Dusp1* null mice present an early imbalance in ROS-detoxifying enzymes (MnSOD) and in glutathione synthesis (*Glcm*) and metabolism (*Gpx1* and *Gsr*) enzymes. A functional imbalance is implied because levels of mRNAs encoding antioxidant enzymes have been shown to correlate with protein levels and enzyme activities (*Iskusnykh et al., 2013*; *Sikalidis et al., 2014*; *Takeshita et al., 2000*). Ageing is associated with increasing oxidation, the progression of cellular damage and cell death, but here, levels mRNAs that encode redox enzymes are altered only in 2-month-old null mice. This situation could be similar to that reported in apolipoprotein E (APOE)-deficient mice, in which a pre-lesion increase in antioxidant enzymes is followed by a decrease once the vascular lesion is extended (*Hoen et al., 2003*).

Taken together, these results suggest the existence of a pro-oxidative stage predisposition in the DUSP1-deficient mouse. Our data point to the activation of MAPK14 as the possible link between the absence of DUSP1 and a premature increase in cochlear oxidative stress and pro-inflammatory status. MAPK14 is expressed in the cochlea (*Parker et al., 2015*; *Sanchez-Calderon et al., 2010*) and it is activated at different time points after noise insult (*Jamesdaniel et al., 2011*; *Maeda et al., 2013*; *Murillo-Cuesta et al., 2015*), as well as during ageing (*Sha et al., 2009*). We show here that DUSP1 is induced by noise-exposure and that its transcription increases with age. MAPK14 inhibitors attenuate mitochondrial and cytoplasmic ROS production in DUSP1-deficient macrophages (*Talwar et al., 2017*). In this sense, the suppression of MAPK14 activity has been linked with protection from mitochondrial dysfunction and ROS generation in different pathologies (*Huang et al., 2018*; *Yu et al., 2017*). Furthermore, the inhibition of MAPK14 downregulates the expression of NADPH oxidase, whereas the activation of MAPK14 induces the phosphorylation and activation of NADPH oxidase subcomponents, potentially leading to enhanced ROS production (*Manea et al., 2015*). These previous sources of evidence are in agreement with our results: *Dusp1* null mice exhibited increased sensitivity to noise exposure, whereas $Mapk14^{KI/KI}$ mice showed better hearing thresholds and better auditory recovery after the insult than wildtype mice, suggesting an important role of MAPK14/DUSP1 in the progression of stress-induced injury.

*Dusp1* deficiency also leads to an early increase in kidney injury molecule-1 (KIM-1), which is induced in the cochlea after cisplatin treatment (*Mukherjea et al., 2006*) and is involved in kidney inflammatory processes that regulate macrophage activation and migration (*Tian et al., 2017*). Chronic inflammation has been associated with the worsening of human presbycusis (*Verschuur et al., 2014*). NOX3-produced ROS is coupled with the production of inflammatory cytokines (*Mukherjea et al., 2011*) and DUSP1 is an important regulator of the innate immune responses (*Chi et al., 2006*). A chronic inflammatory state that occurs as a consequence of ageing of the immune system (known as 'inflammaging') is another hallmark of cochlear ageing (*Kalinec et al., 2017*). In this context, *Dusp1* null mice present an exacerbated inflammatory response from the age of 5 months onwards. Our results suggest that an inflammatory feedback loop may exist in which the presence of inflammatory cytokines, the absence of DUSP1 and the consequent activation of stress kinases enhances cellular apoptosis.

Null mice showed increased expression of cytokines and of IBA1-positive phagocytic cells from the youngest age studied. Interestingly, during normal ageing, resident macrophages increase in number and undergo changes in morphology that are directly associated with the degeneration of sensorial cells (*Frye et al., 2017*). Chronic inflammation causes damage to cells that are quickly

removed from the cochlea (*Hu et al., 2002*). Resident macrophages and supporting cells of the organ of Corti have been proposed as the effectors of the cleaning of cellular debris (*Anttonen et al., 2014*; *Hirose et al., 2005*).

Taken together, our results show that *Dusp1* deficiency accelerates the onset and the progression of age-related hearing loss. Our results therefore suggest that DUSP1 is essential for lifelong cochlear homeostasis.

# Materials and methods

**Key resources table**

| Reagent type (species) or resource | Designation | Source or reference | Identifiers | Additional information |
|---|---|---|---|---|
| Genetic reagent (Mus musculus) | *Dusp1*$^{-/-}$ | *Dorfman et al., 1996* | RRID:MGI:4940296 | R Perona lab (Institute of Biomedical Research 'Alberto Sols', Madrid, Spain) |
| Genetic reagent (M. musculus) | *Mapk14*$^{+/KI}$; *Mapk14*$^{KI/KI}$ | *Ventura et al., 2007* | RRID:MGI:3716853 | AR Nebreda lab (Institute for Research in Biomedicine, Barcelona, Spain) |
| Cell line (M. musculus) | *Dusp1*$^{-/-}$ MEF | *Dorfman et al., 1996* | RRID:MGI:4940296 | Primary culture of mouse embryonic fibroblasts, maintained in I Sánchez-Pérez's lab |
| Antibody | Anti-myelin P0 (chicken polyclonal) | Neuromics | Neuromics Cat# CH23009; RRID:AB_2737144 | IHC (1:150) |
| Antibody | Anti-nitrotyrosine (rabbit polyclonal) | Merck-Millipore | Millipore Cat# AB5411; RRID:AB_177459 | IHC (1:100) |
| Antibody | Anti-IBA1 (goat polyclonal) | Abcam | Abcam Cat# ab5076; RRID:AB_2224402 | IHC (1:100) |
| Antibody | Anti-MyoVIIA (rabbit polyclonal) | Proteus | Proteus Biosciences Cat# 25–6790; RRID:AB_2314838 | IHC (1:250) |
| Antibody | Anti-neurofilament (mouse monoclonal) | Merck-Millipore | Millipore Cat# CBL212; RRID:AB_93408 | IHC (1:100) |
| Antibody | Alexa Fluor 647 Phalloidin | Thermo Fisher Scientific | Thermo Fisher Scientific Cat# A22287; RRID:AB_2620155 | IHC (1:1000) |
| Antibody | Anti-P-p38 (rabbit polyclonal) | Cell Signaling | Cell Signaling Technology Cat# 9211, RRID:AB_331641 | WB (1:1000; 1:2000) |
| Antibody | Anti-P-JNK (rabbit polyclonal) | Cell Signaling | Cell Signaling Technology Cat# 4668; RRID:AB_823588 | WB (1:1000) |
| Antibody | Anti-P-ERK (rabbit polyclonal) | Cell Signaling | Cell Signaling Technology Cat# 9101, RRID:AB_331646 | WB (1:1000) |
| Antibody | Anti-MKP1 (C-19) (rabbit polyclonal) | Santa Cruz Biotechnology | Santa Cruz Biotechnology Cat# sc-370; RRID:AB_631385 | WB (1:1000) |
| Antibody | Anti-γ-H2AX Ser139 (rabbit polyclonal) | Cell Signaling | Cell Signaling Technology Cat# 2577; RRID:AB_2118010 | IHC (1:200); WB (1:1000) |

*Continued on next page*

*Continued*

| Reagent type (species) or resource | Designation | Source or reference | Identifiers | Additional information |
|---|---|---|---|---|
| Antibody | Anti-P22phox (rabbit polyclonal) | Santa Cruz Biotechnology | Santa Cruz Biotechnology Cat# sc-20781; RRID:AB_2090309 | WB (1:250) |
| Antibody | Anti-MnSOD (rabbit polyclonal) | Merck-Millipore | Millipore Cat# 06–984; RRID:AB_310325 | WB (1:1000) |
| Antibody | Anti-BAX (NT) (rabbit polyclonal) | Merck-Millipore | Millipore Cat# ABC11; RRID:AB_10561771 | WB (1:1000) |
| Antibody | Anti-PI3K (rabbit polyclonal) | Not commercially available | | WB (1:10,000) From AM Valverde's lab (Institute of Biomedical Research 'Alberto Sols', Madrid, Spain) |
| Antibody | Anti-p38 (C-20) (rabbit polyclonal) | Santa Cruz Biotechnology | Santa Cruz Biotechnology Cat# sc-535; RRID:AB_632138 | WB (1:1000) |
| Antibody | Anti-caspasa3 (rabbit polyclonal) | Cell Signaling | Cell Signaling Technology Cat# 9662; RRID:AB_331439 | WB (1:1000) |
| Antibody | Anti-β-actin (mouse monoclonal) | Sigma-Aldrich | Sigma-Aldrich Cat# A5441; RRID:AB_4767441 | WB (1:10000) |
| Antibody | Anti-β-tubulin (mouse monoclonal) | Sigma-Aldrich | Sigma-Aldrich Cat# T9026; RRID:AB_47759 | WB (1:1000) |
| Commercial assay or kit | Dead-End Fluorometric TUNEL System | Promega | Promega Cat# G3250 | |
| Chemical compound, drug | TNFα | Sigma-Aldrich | Sigma-Aldrich Cat# T6674 | (10 ng/ml) |
| Software, algorithm | BioSigRP TM | Tucker Davis Technologies (TDT) | RRID:SCR_014590 | |
| Software, algorithm | Fiji | Fiji (https://fiji.sc/) | RRID:SCR_002285 | |
| Software, algorithm | SPSS | IBM | RRID:SCR_002865 | |
| Software, algorithm | Cell Profiler | Cell Profiler (https://cellprofiler.org/) | RRID:SCR_007358 | |

## Mice

129S2/SvPas:C57BL/6 wildtype ($Dusp1^{+/+}$), heterozygous ($Dusp1^{+/-}$) and null ($Dusp1^{-/-}$) mice were generated and genotyped as reported (*Dorfman et al., 1996*). $Dusp1^{-/-}$ mice are fertile and do not show higher mortality rates than wildtypes. General phenotype data for *Dusp1* knockout mice can be found in the Mouse Genome Database (MGD), Mouse Genome Informatics (MGI: 4940244; http://www.informatics.jax.org). Hearing was studied monthly in several cohorts and then data-grouped afterwards into groups of 2, 4–5, 8–9, 12 and 16–19 months. Certain cohorts were used to obtain specific time points for further analysis. Neither male and female wildtype mice nor male and female null mice showed sex-associated differences in auditory phenotype. When indicated, some experiments were performed in wildtype mice with a mixed genetic background of 129/SvEvTac (Taconic) and HsdOla:MF1 (Harlan Laboratories) (129Sv:MF1).

The *Mapk14* knockout mouse is embryo lethal, therefore the conditional knockin mouse $Mapk14^{tm2Nbr}/Mapk14^{tm2Nbr}$ $Polr2a^{tm1(cre/ERT2)Bbd}/Polr2a^+$ ($p38^{flox/flox}$ $Polr2a^{KI/+}$ (MGI 3716853) was used (*Ventura et al., 2007*). Male and female $p38^{flox/flox}$ $Polr2a^{KI/+}$ were matted to obtain $p38^{flox/flox}$ $Polr2a^{KI/+}$ ($Mapk14^{+/KI}$), $p38^{flox/flox}$ $Polr2a^{KI/KI}$ ($Mapk14^{KI/KI}$) and $p38^{flox/flox}$ $Polr2a^{+/+}$ ($Mapk14^{+/+}$) littermates. One-month-old mice from the three genotypes were treated intraperitoneally with 4-hydroxytamoxifen (Tamoxifen (TAM), T5648, Sigma-Aldrich, St. Louis, MO, USA, in corn oil, 75 mg/kg/day) or corn oil (C8267, Sigma-Aldrich), 3.75 ml/g/day, for 5 days, following the JAX TAM

injection protocol for inducible Cre-derived lines. This protocol achieves the excision of exons 2 and 3 of the gene coding for p38α (*Mapk14*) with high efficiency (85%–95%).

Animal experimentation was conducted in accordance with European Community 2010/63/EU and Spanish RD 53/2013 guidelines.

## Cell culture

Mouse embryonic fibroblasts (MEFs) were prepared from embryos of the different genotypes at day 13.5 of gestation as previously described (*Palmero and Serrano, 2001*). After disaggregation of embryos and brief expansion, MEFs from individual embryos were stored in liquid nitrogen until use. Plating of MEFs after thawing was considered passage 1. Cells were maintained in Dulbecco's Modified Eagle Medium (DMEM) containing 10% fetal bovine serum and a cocktail of antibiotics and anti-mycotics in a $CO_2$ incubator at 37°C. MEFs prepared from wildtype and null mice were treated with TNFα (10 ng/ml) (T6674, Sigma-Aldrich) at the times indicated.

## Hearing evaluation

Mice were anesthetized with a mixture of ketamine (100 mg/kg; Imalgene 1000, Merial, Lyon, France) and xylazine (10 mg/kg; Rompun 2%, Bayer, Leverkusen, Germany). Auditory brainstem responses (ABR) were measured using a Tucker Davis Technologies workstation (Tucker Davis Technologies, Alachua, FL, USA). Electrical responses were measured in response to broadband click and 8, 16, 20, 28 and 40 kHz pure tone stimuli, with an intensity range 90–20 dB SPL in 5–10 dB steps, as previously reported (*Cediel et al., 2006*). Peak and interpeak latencies were analyzed at 80 dB SPL above hearing threshold after click stimulation. Recording of distortion product otoacoustic emissions (DPOAEs) was performed after stimulation with f1 and f2 primary tones, with a ratio f2/f1 = 1.2, using a TDT equipment, as described (*Martínez-Vega et al., 2015b*). Primary tones for 8, 10 and 14 kHz frequencies were tested. Analyses of waves, thresholds and latencies were performed with BioSigRP TM software (Tucker Davis Technologies). ABR registries were taken in different cohorts from the age of 1 month.

## Noise exposure

To evaluate the susceptibility to noise, 2-month-old mice from the three genotypes (*Dusp1* mutants, *Mapk14* mutants and wildtype) were exposed to violet swept sine noise at 107 dB SPL for 30 min. The effect of noise exposure on hearing was evaluated by ABR before (baseline) and 3, 14 and 28 days after exposure.

## Middle and inner ear dissections and morphological evaluation

Two-month-old mice of each genotype were administered a lethal dose of pentobarbital (Dolethal, Vétoquinol, Madrid, Spain). The inner ear and the ossicles — the malleus, incus, and stapes — of the middle ear were dissected as reported (*Sakamoto et al., 2017*). Microphotographs of these structures were taken using a digital camera connected to a Leica MZ8 stereo microscope (Leica, Wetzlar, Germany).

## Cochlear morphology and immunohistochemistry

For histological analysis, mice were injected with a pentobarbital overdose and perfused with PBS/paraformaldehyde as previously described (*Camarero et al., 2001*; *Sanchez-Calderon et al., 2010*). Cochleae were then dissected, postfixed and decalcified before being embedded in paraffin or Tissue-Tek OCT (Sakura Finetek, Torrance, CA, USA). Paraffin cochlear sections (7 µm) were either stained with hematoxylin/eosin or used for immunohistochemical investigations with anti-myelin P0 (chicken, 1:150, CH23009, Neuromics, Edina, MN, USA) and anti-nitrotyrosine (rabbit, 1:100, AB5411, Merck-Millipore, Burlington, MA, USA) (*Martínez-Vega et al., 2015a*). Representative images were taken with a Zeiss microscope connected to a DP70 digital camera (Olympus, Tokyo, Japan).

For immunofluorescence assays, frozen OCT cochlear sections (10 µm) were treated overnight, as described previously (*Martínez-Vega et al., 2015a*), with anti-IBA1 (goat, 1:100, ab5076, Abcam, Cambridge, UK). Fluorescent images were taken with an epifluorescence (Nikon 90i, Tokyo, Japan) microscope. Analyses of total IBA1 intensity and 3-NT mean gray level were performed with Fiji

software v1.51n (*Schindelin et al., 2012*). We analyzed the spiral ligament of each cochlear turn in four serial cryosections per animal (preparing at least three mice of each genotype of 4–5 months of age and 8–9 months of age) or the spiral ganglia of the basal and middle turns in four serial paraffin sections per animal (prepared from at least three 2-month-old mice of each genotype).

MEFs cells were fixed in 4% formaldehyde for 20 min, washed with PBS and permeabilized with Triton 0.5% for 10 min, before blocking with BSA 5% for 1 hr. Samples were incubated overnight with the primary antibody $\gamma$-H2AX Ser$^{139}$ (rabbit, 1:200, Merck-Millipore) at 4°C, followed by a 1 hr incubation with the adequate secondary antibody (1:500, Alexa Fluor 488, Invitrogen) at room temperature. DNA was stained with DAPI. Fluorescence microscopy was performed using a NIKON Eclipse 90i, and the software programs Nikon NIS-Elements and Image J were used for image analysis. $\gamma$-H2AX foci were quantified with Cell Profiler software and analyzed with IBM SPSS v.22, which was used to perform two-way ANOVA tests.

## TUNEL assay

Apoptosis was evaluated by TdT-mediated dUTP nick-end labeling (TUNEL) using the Dead-End Fluorometric TUNEL System (Promega, Madison, WI, USA), essentially as described by the manufacturer. Deparaffinized sections were postfixed using 4% PFA (pH 7.4) before and after treatment with proteinase K. Sections were incubated with the TdT enzyme for 1 hr and mounted with Vectashield mounting medium with DAPI before visualizing in a Nikon 90i microscope. TUNEL-positive cells were counted in the spiral ganglia in four serial paraffin sections per animal (prepared from at least three 4–5-month-old mice of each genotype) using Fiji software. Representative fluorescent stack images of the middle and basal cochlear turns were taken using a confocal microscope (LSM710 Zeiss, Oberkochen, Germany) with a glycerol-immersion objective (63x).

## Organ of Corti dissection, cochleogram and hair cell quantification

The organ of Corti of decalcified cochleae was dissected into half-turns. The pieces were permeabilized with 1% Triton X-100 (Merck Millipore, Billerica, MA, USA), blocked with 5% normal goat serum (Sigma-Aldrich) and incubated overnight at 4°C with anti-MyoVIIA (rabbit, 1:250, PT-25–6790, Proteus, Ramona, CA, USA) or anti-neurofilament (mouse, 1:100, CBL212, Merck-Millipore). Alexa Fluor secondary antibodies or Alexa Fluor 647 Phalloidin (1:1000, A22287, Thermo Fisher Scientific, Waltham, MA, USA) were incubated at 1:200 for 2 hr at RT. The half-turns were then incubated with DAPI (1:1000, Thermo Fisher Scientific) and mounted with Prolong (Thermo Fisher Scientific), and low magnification fluorescent images were taken with Nikon 90i microscope. The cochleogram was plotted using a custom Fiji plugin as reported previously (plugin freely available at http://www.mas-seyeandear.org/research/otolaryngology/investigators/laboratories/eaton-peabody-laboratories/epl-histology-resources/). The numbers of inner (IHC) and outer (OHC) hair cells were counted using Fiji software in 200 µm sections in the apical, middle and basal regions, which are defined as grouped percentages of distance from apex (apical <21%, middle 21–47%, basal >47%). Segments were prepared from four mice of each genotype. Representative fluorescent stack images were taken at specified cochlear regions using a confocal microscope (LSM710 Zeiss, Oberkochen, Germany) with a glycerol-immersion objective (63x) and image spacing in the z plane of 0.7 µm.

## RT-qPCR

Cochlear RNAs were extracted using an RNeasy kit (QIAGEN, Hilden, Germany) and the quality and quantity of these RNAs were determined using an Agilent Bioanalyzer 2100 (Agilent Technologies Santa Clara, CA, USA). RNA pools were made from at least three different animals from each age group and genotype. cDNA was then generated by reverse transcription (High Capacity cDNA Reverse Transcription Kit; Applied Biosystems, Foster City, CA, USA) and gene expression was analyzed in triplicate by qPCR on Applied Biosystems 7900 HT using TaqMan Gene Expression Assays (Applied Biosystems) for *Gpx1*, *Gpx4*, *Gsr*, *Gss*, *Gclc*, *Gclm*, *Cbs*, *Nox3*, *Nox4*, *Ucp1*, *Catalase*, *Cyba*, *Kim1*, *Apaf1*, *Il1b*, *Il6*, *Tnfa*, *Tgfb1*, *Il10*, *Foxp3*, *Mpz*, *RbFox3*, *Sox2* and *Prestin* (*Supplementary file 2*). In addition, cDNAs were amplified in triplicate using gene-specific primers (*Supplementary file 3*) and Power SYBR Green PCR Master Mix (Applied Biosystems) to evaluate the expression of MAP kinase phosphatases. *Rplp0* or *Hprt1* were used as endogenous

housekeeping control genes and the estimated gene expression was calculated as $2^{-\Delta\Delta Ct}$, as reported previously (*Sanchez-Calderon et al., 2010*).

## Western blotting

Cochlear proteins from 4-month-old mice (a pool of three cochleae of each genotype) were extracted using the extraction buffer of a Ready Protein Extraction Kit (Bio-Rad, Hercules, CA, USA) fsupplemented with 0.01% protease and 0.01% phosphatase inhibitors (Sigma-Aldrich).

Equal volumes of cochlear proteins were resolved using denaturing SDS-PAGE, followed by transfer to PVDF (Bio-Rad) membranes using Bio-Rad Trans Blot TURBO apparatus. Membranes were blocked using 5% BSA or non-fat dried milk in 0.075% Tween-TBS 1 mM and incubated overnight with the following antibodies: rabbit anti-P-p38 (1:1000, 9211, Cell Signaling, Danvers, MA, USA), rabbit anti-P-JNK (1:1000, 4668, Cell Signaling), rabbit anti-P-ERK (1:1000, 9101, Cell Signaling), rabbit anti-MKP1 (C-19) (1:1000, sc-370, Santa Cruz, Biotechnology), rabbit anti- $\gamma$-H2AX Ser$^{139}$ (1:1000, 2577, Cell Signaling), rabbit anti-P22phox (1:250, sc-20781, Santa Cruz, Biotechnology, Dallas, TX, USA), rabbit anti-MnSOD (1:1000, 06–984, Merck-Millipore), rabbit anti-BAX (NT) (1:1000, ABC11, Merck-Millipore), and rabbit anti-PI3K (1:10,000). Immunoreactive bands were visualized using Clarity TM Western ECL Substrate (Bio-Rad) with an ImageQuant LAS4000 mini digital camera (GE Healthcare Bio-Sciences). Densities of the immunoreactive bands were quantified by densitometry using ImageQuant TL software.

Total protein extracts (WCE) were obtained from MEFs using the previously described lysis buffer (*Sánchez-Perez et al., 1998*). Twenty micrograms of WCE per sample were loaded onto 10% SDS-PAGE polyacrylamide gels, and then transferred onto nitrocellulose membranes, or onto PVDF (Bio-Rad) membranes using Bio-Rad Trans Blot TURBO apparatus, followed by immunodetection using appropriate antibodies. Antibodies against the following proteins were: rabbit anti-MKP1 (C-19) (1;1000, sc-370, Santa Cruz, Biotechnology), rabbit anti P-p38 (1:2000, #9211, Cell Signaling), rabbit anti-p38 (C-20) (1:1000, sc-535, Santa Cruz, Biotechnology), rabbit anti-caspase3 (1:1000, #9662, Cell Signaling), mouse anti-β-actin (A5441, Sigma Aldrich) and mouse anti-β-tubulin as the loading control (T9026, Sigma Aldrich).

## ROS levels

Cellular ROS levels were quantified by staining adherent cells with DHE followed by cell lifting with trypsin-EDTA and flow cytometric analysis. To quantify ROS by flow cytometry, adherent cells were stained with DHE for 45 min in PBS at 37°C in the dark. DHE is oxidizedto ethidium by superoxide in live cells (*Du et al., 2009*). Cells were lifted with trypsin-EDTA and washed, before fluorescence data were acquired within 60 min using the BD FACS Scan with a 488 nm laser and 585 nm DHE bandpass filters.

## Quantification of DNA damage

Cells were fixed in formaldehyde for 20 min, washed with PBS and permeabilized with Triton 0.5% for 10 min, before blocking with BSA 5% for 1 hr. Samples were incubated overnight with the primary antibody $\gamma$-H2AX Ser$^{139}$ (Millipore) at 4°C, followed by a 1 hr incubation with the adequate secondary antibody at room temperature. DNA was stained with DAPI. Fluorescence microscopy was performed using a NIKON Eclipse 90i, and for the image analysis, the software program Nikon NIS-Elements and Image J were used. Secondary antibodies, conjugated with Alexa Fluor 488 (1:500), were purchased from Invitrogen. $\gamma$-H2AX foci were quantified with Cell Profiler software and analyzed using IBM SPSS 22 to perform two-way ANOVA tests.

## Statistical analysis

Unless otherwise specified, data were analyzed by Student T-test after a Levene's or Fisher test of equality of variances with SPSS v23.0 software. Sample size was estimated to obtain a 90% statistical power with a 0.05 significance level, using data from previous experiments and calculating the Cohen's d value. Data are expressed as mean ± SEM. The results were considered significant at $p < 0.05$.

## Acknowledgments

This work has been supported by Spanish MINECO/FEDER SAF2017-86107-R to IVN and P17-01401 (Fondo de Investigaciones Sanitarias, Instituto de Salud Carlos III, Spain) supported by FEDER funds to RP and IS-P. SM, and LR hold CIBER ISCIII researcher contracts. AMC was supported by contracts from FP7-PEOPLE-2013-IAPP TARGEAR and FEDER/CIBERER. We warmly thank Prof. Angel Nebreda (Institute for Research in Biomedicine, Barcelona, Spain) for the kind gift of the *Mapk14* conditional mouse and for advice on the procedures used.

## Additional information

### Funding

| Funder | Grant reference number | Author |
| --- | --- | --- |
| Ministerio de Economía y Competitividad | SAF2017-86107-R | Isabel Varela-Nieto |
| Instituto de Salud Carlos III | P17-01401 | Rosario Perona Isabel Sánchez-Pérez |

The funders had no role in study design, data collection and interpretation, or the decision to submit the work for publication.

### Author contributions

Adelaida M Celaya, Data curation, Formal analysis, Supervision, Investigation, Visualization, Writing—original draft; Isabel Sánchez-Pérez, Jose M Bermúdez-Muñoz, Data curation, Formal analysis, Investigation, Visualization, Writing—review and editing; Lourdes Rodríguez-de la Rosa, Silvia Murillo-Cuesta, Data curation, Formal analysis, Investigation; Laura Pintado-Berninches, Resources, Methodology, Given final approval of the version to be published; Rosario Perona, Resources, Supervision, Funding acquisition, Writing—review and editing; Isabel Varela-Nieto, Conceptualization, Formal analysis, Supervision, Funding acquisition, Visualization, Writing—original draft, Project administration

### Author ORCIDs

Adelaida M Celaya (iD) http://orcid.org/0000-0002-0757-6163
Isabel Sánchez-Pérez (iD) https://orcid.org/0000-0002-4829-201X
Jose M Bermúdez-Muñoz (iD) http://orcid.org/0000-0002-6034-9285
Silvia Murillo-Cuesta (iD) http://orcid.org/0000-0002-8706-4327
Isabel Varela-Nieto (iD) https://orcid.org/0000-0003-3077-0500

### Ethics

Animal experimentation: Animal experimentation was conducted in accordance with Spanish (RD 53/2013) and European (Directive 2010/63/EU) legislations. All protocols used in this study were reviewed and approved by the Ethical Committee of Animal Experimentation at IIBm and Ethical Committee at CSIC in a facility inscribed in the official registration of breeding establishments, suppliers and users of experimental animals in the Ministry of Agriculture, Fisheries and Food (registration number, ES280790000188). Mice procedures were done according with scientific, humane, and ethical principles. The studied mouse model did not show phenotype differences comparing male and female. Thus, to ensure that our research represents both genders, the studies describes in this work were performed using both sexes equitably. The number of biological and experimental replicates is detailed in the legend of each figure.

### Decision letter and Author response

Decision letter https://doi.org/10.7554/eLife.39159.023
Author response https://doi.org/10.7554/eLife.39159.024

## Additional files

### Supplementary files

• Supplementary file 1. Monthly ABR threshold (dB SPL).
DOI: https://doi.org/10.7554/eLife.39159.016

• Supplementary file 2. Taqman essays for RT-qPCR.
DOI: https://doi.org/10.7554/eLife.39159.017

• Supplementary file 3. Primers for RT-qPCR. Primers for MKPs were designed using Primer Express 3.0 software and the mouse gene sequences available on the Ensembl genome database with references: NM_013642.3 (*Dusp1*), NM_010090.2 (*Dusp2*), NM_176933.4 (*Dusp4*), NM_001085390.1 (*Dusp5*), NM_026268.3 (*Dusp6*), NM_153459.4 (*Dusp7*), NM_008748.3 (*Dusp8*), NM_022019.6 (*Dusp10*) and NM_130447.3 (*Dusp16*). Base numbers indicate the location of the primer sequences in the corresponding mRNA; primers for *Dusp1* were designed in the region of exon 2.
DOI: https://doi.org/10.7554/eLife.39159.018

• Transparent reporting form
DOI: https://doi.org/10.7554/eLife.39159.019

### Data availability

Source data files have been provided for ABR data in Figures 2 and Figure2-figure supplement 1, as well as for gene expression data in Figures 1, 3, 5, 6 and Figure1-figure supplement 1. Data has also been deposited on Dryad under the doi: 10.5061/dryad.51m8c58.

The following dataset was generated:

| Author(s) | Year | Dataset title | Dataset URL | Database and Identifier |
|---|---|---|---|---|
| Celaya AM, Sánchez-Pérez I | 2019 | Data from: Deficit of mitogen-activated protein kinase phosphatase 1 (MKP1) accelerates progressive hearing loss | http://dx.doi.org/10.5061/dryad.51m8c58 | Dryad Digital Repository, 10.5061/dryad.51m8c58 |

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
