## [Decision Letter]

[Editors’ note: this article was originally rejected after discussions between the reviewers, but the authors were invited to resubmit after an appeal against the decision.]

Thank you for submitting your work entitled "Deficit of mitogen-activated protein kinase phosphatase 1 (MKP1) accelerates progressive hearing loss" for consideration by *eLife*. Your article has been reviewed by a Senior Editor, a Reviewing Editor, and two reviewers. The following individuals involved in review of your submission have agreed to reveal their identity: Sally Dawson (Reviewer #2).

Our decision has been reached after consultation between the reviewers. Based on these discussions and the individual reviews below, we regret to inform you that your work will not be considered further for publication in *eLife*.

Please accept our apologies for the long delay in the reviewing of your manuscript, due to the summer holidays. The reviewers found your work interesting, but raised several issues and concerns. It is not, therefore, possible to accept your manuscript for publication in its current state, and the changes required are deemed to be too extensive to be addressed within the window we normally allow for revisions. Should you wish to carry out these revisions and send a revised version of your paper to *eLife*, it will be treated as a new submission.

*Reviewer #2:*

Celaya et al., present an analysis and characterisation of an auditory phenotype in a mouse model with targeted deletion of the MKP1 gene. Their extensive analysis includes expression analysis of MKP1 gene by RT-qPCR, evidence of auditory functional deficit by ABR and DPOAEs, and a quantitative assessment of hair cell and neuronal cellular loss in the cochlea which is correlated with the auditory deficit. Furthermore, they investigate the mechanism underlying the hearing loss and find some evidence for effects on reduced resistance to oxidative stress. I have very little in the way of criticism of what is included in the manuscript, which I find to be carried out well, clearly presented and proportionately discussed within the text.

My comments mainly refer to omissions from the manuscript that would improve the manuscript and relatively minor clarifications:

1) No mention is made of the systematic effects of MKP1 deletion. What are these and could they have indirect on the auditory phenotype?

2) The authors do not give an indication of whether there is a phenotype in the heterozygote mice and do not present any data from heterozygotes. This is disappointing; I would not expect a full characterisation of the heterozygous mice (hets) but it is important to know whether there is a hearing deficit in the hets. The authors suggest that MKP1 may play a role in general ARHL risk and pathogenesis and therefore this would be strengthened if 50% MKP1 also causes an auditory deficit. The manuscript would therefore be strengthened with addition of ABR data from hets.

3) The approved name for this gene/protein is now DUSP1 which should be used within the manuscript, rather than MKP1.

4) Figure 4, if available a lower zoom image of the wholemounts would provide an indication of base to apex gradient in hair cell loss.

5) RT-qPCR data although it gives a quantitative assessment of cochlear gene expression it does not take into account the many cell types in the cochlea and an immunofluorescence characterisation of MKP1 and the oxidative stress genes would provide cell specific expression data. Have the authors attempted an immuno-characterisation of MKP1 expression?

6) Could they clarify when SYBR green assays were used and when Taqman assays were used by giving Taqman assay IDs. Why were two different internal housekeeping gene assays used?

7) It would be interesting to know whether these mice are at increased risk of noise induced hearing loss, although probably not required for publication.

8) Although mostly well written there are some typos and grammatical errors particularly in the abstract which I suspect may have had some last minute changes, which were not proofread. ARHL is quite often mis-written as AHRL throughout manuscript.

9) Discussion section, authors state "oxidative stress as main cause of later problems". Since they have not investigated any other mechanism then this is an over-statement.

*Reviewer #3:*

The authors claimed that Mkp1 expression is age-regulated in the mouse cochlea and Mkp1 gene knock-out caused premature progressive hearing loss, suggesting that MKP1 is essential for cochlear homeostasis during ageing by regulating cochlear oxidative stress and inflammation.

The capability of MKP1 to regulate oxidative/inflammatory status in the cochlea is innovative and appealing topic. However, the principal limitation of the study is that the Authors omitted to describe/speculate on the molecular mechanism that link MKP1 deficiency with the increased oxidative stress/inflammation in the cochlea. Furthermore, there are a lot of criticisms that need to be carefully addressed.

Essential revisions:

1) The criteria of inclusion of the animals in this study is not clear and this is a crucial point considering that the paper is focused on the age related hearing loss. Namely, it should be explained why the range of age for animal inclusion is wide: 2, 4-5, 8-9, 12 and 16-19 months of age. Considering the lifespan of this mice the differences of 2-3 months are relevant, especially in the old animals. Furthermore, the differences on the phenotype and the auditory function assessment should be illustrated for each group even if major data have been provided in 2-5-8 months of age group.

2) Hearing function was assessed from 2 to 12 month-old and MKP1^-/-^ mice show ABR thresholds significantly higher with respect to controls (both for click and tone burst responses in almost all frequencies analyzed) already at 2 months of age; then, hearing function worsened with age compared to MKP1^+/+^ mice. It could be useful to assess hearing thresholds in an earlier time point, in order to demonstrate that MKP1^-/-^ mice show baseline threshold values similarly to controls and that hearing worsen with time compared to MKP1^+/+^ mice. This is relevant issue because the authors state that MKP1 deficiency accelerates the onset and the progression of age-related hearing loss. While, but, based on the current electrophysiological data, it is not clear if MKP1 deficiency also impairs the development of auditory function.

3) It is not clear how DPOAE was recorded. In the Materials and methods section the authors refer to a previous published procedure (Martinez-Vega et al., 2015) but in this paper DPOAE recording was not performed. Also, OHC count reveals a marked decrease of cell number in the basal turn of MKP1^-/-^ mice, however DPOAE recording was performed by using primary tones for 8-14 kHz. Maybe, it could be useful to analyze also responses for higher frequencies. Also, it is not clear why DP level increased from 4-5 to 8-9 months, when the level of *Prestin* decreased (as shown in Figure 3B). I would expect a decrease of DPOAE responses in MKP1^-/-^ mice rather than an increase.

4) The authors analyzed the level of genes involved in oxidative stress in cochlear samples from both genotypes and the ROS expression (DHE assay) was studied in MEF cells derived from MKP1^-/-^ and MKP1^+/+^ mice. Although the methodology was well conducted, in order to support the major conclusion that MKP1 is essential for the regulation of cochlear redox balance, it could be relevant to establish a direct relationship between cochlear oxidative stress and MKP1 activation in vivo. It could be helpful to perform some quantification (i.e. western blot analyses) on cochlear samples from MKP1^-/-^ and MKP1^+/+^ mice in order to correlate the modifications in cochlear redox status during aging in both genotypes with MKP1 level expression.

4) Morphological observations have been performed only in 4-8 months old mice, however, ABRs revealed hearing loss also at 2 months of age in MKP1^-/-^ mice, thus morphological data at earlier time points could be added to support electrophysiological data.

6) Figure 4: The quality of panels a-h could be ameliorated. Maybe fluorescence excitation was saturated, and this does not allow to appreciate the morphological details of the organ of Corti. Also, I suggest modifying DPOAE data presentation. Maybe it could be useful for data interpretation to graph DPOAEs in response to increasing stimulus intensity (showing 2f1-f2 dB SPL in y axis and f1 and f2 input level in the x axis), selecting only one representative frequency.

[Editors’ note: what now follows is the decision letter after the authors submitted for further consideration.]

Thank you for choosing to send your work entitled "Deficit of mitogen-activated protein kinase phosphatase 1 (MKP1) accelerates progressive hearing loss" for consideration at *eLife*. Your letter of appeal has been considered by a Senior Editor, a Reviewing editor and two reviewers, and we are prepared to consider a revised submission with no guarantees of acceptance.

The reviewers differ in their opinions as to whether you will be able to perform the experiments required to provide new evidence concerning the role of MKP1 in the molecular mechanism and to demonstrate the link you propose between MKP1 and p38 activation, within the two months allowed. We are, however, willing to give you the opportunity to try to do this and advise you to pay close attention to the comments of reviewer 3.

Reviewer #3:

Point 1:

In my opinion this criticism remains unsolved in absence of adjunctive experiments. The authors can improve the interpretation of results hypothesizing putative molecular mechanisms leading to activation of oxidative stress/inflammatory pathways, however, providing new evidence on the role of MKP1 molecular mechanism is challenging. The authors suggest a link between MKP1 and p38 activation that could be studied.

Point 2:

I am not in agreement with this experimental plan, because in my opinion, given the aging processes are one of the major topic of this study, more careful samples of animal age should be done. However, at least I suggest clarifying this point in the manuscript and to report this comment in Discussion section.

Point 3:

I suggest including data regarding ABR recordings at P31. However, considering that the authors state that at 2 months cell types are formed and there are no obvious morphological alterations or cytoarchitecture disorganization, even more so I think that they have to explain the threshold elevation observed in 2 months MKP1 KO animals with respect to Wt age-matched animals.

Point 4

I agree, these data have to be added to improve the study relevance.

Point 5

I agree, however, on the basis of these morphological data, as mentioned above, the Authors must to explain why the KO animals show a threshold elevation compared to WT controls at 2 months age. More details could be added.

---

## [Author Response]

[Editors’ note: the author responses to the first round of peer review follow.]

Reviewer #2:[…] My comments mainly refer to omissions from the manuscript that would improve the manuscript and relatively minor clarifications:

We appreciate very much the interesting comments and positive input by this reviewer.

1) No mention is made of the systematic effects of MKP1 deletion. What are these and could they have indirect on the auditory phenotype?

In our experience, *Dusp1*mice are fertile and do not show higher mortality rates than wild types, or any other obvious general phenotypic trait.

Furthermore, *Dusp1*knock out mice have been studied and the general phenotype described in http://www.informatics.jax.org/allele/genoview/MGI:4940244?counter=4.

“Briefly, *Mkp1^-/-^* mice present an exacerbated inflammatory response in LPS-treated mice, with increased circulating levels of IL10, Il16 and TNF. *Mkp1^-/-^* mice are resistant to diet-induced obesity due to enhanced energy expenditure but succumb to glucose intolerance on a high fat diet. They present decreased body weight and decreased susceptibility to age-related obesity, accompanied by a smaller liver in aged mice. Mitochondria of their skeletal muscles exhibit increased respiration. *Mkp1^-/-^* mice are also resistant to stress-induced anhedonia (Dorfman et al., 1996; Hammer et al., 2006; Salojin et al., 2006; Shen et al., 2016; Shi et al., 2010)”.

Up to our knowledge, there are not previous studies focused on ARHL.

The revised article we have includes under the Matherial and Methods section a reference to the aforementioned general phenotype of the null *Dusp1* mice described in www.informatics.jax.org.

2) The authors do not give an indication of whether there is a phenotype in the heterozygote mice and do not present any data from heterozygotes. This is disappointing; I would not expect a full characterisation of the heterozygous mice (hets) but it is important to know whether there is a hearing deficit in the hets. The authors suggest that MKP1 may play a role in general ARHL risk and pathogenesis and therefore this would be strengthened if 50% MKP1 also causes an auditory deficit. The manuscript would therefore be strengthened with addition of ABR data from hets.

The comment is right and indeed we have studied the heterozygous mice and we could have mentioned the data in the original manuscript, however it was already very long and included much information and we decided not to. The heterozygous phenotype observed is presented in Author response image 1 (pale grey dots) and compared with that of wild type and KO mice. Hearing evolution was followed up in a longitudinal experiment up to the age of 9 months. There were no differences in ABR thresholds and, therefore, we discontinued the experiment. These results can be explained by the presence and upregulation of other family members, among other hypothesis.

**Author response image 1. respfig1:** Comparative longitudinal hearing evaluation of *Dusp1^+/+^, Dusp1^+/-^*and *Dusp1^-/-^*mice. Evolution of ABR thresholds (mean ± SEM) in response to click and tone bursts stimuli in *Dusp1^+/+^*(white circles), *Dusp1^+/-^*(grey circles) and *Dusp1^-/-^*(black circles) mice of 2 (*Dusp1^+/+^,* n=10; *Dusp1^+/-^*, n=7; *Dusp1^-/-^*, n=7), 4-5 (*Dusp1^+/+^,* n=28; *Dusp1^+/-^*, n=7; *Dusp1^-/-^*, n=24), 8-9 months of age (*Dusp1^++/+^,* n=24; *Dusp1^+/-^*, n=6; *Dusp1^-/-^*, n=22) and 12 months of age (*Dusp1^+/+^,* n=7; *Dusp1^+/-^*, n=3). Statistically significant differences were analysed by Student’s t-test comparing genotypes, * *Dusp1^+/-^*vs. *Dusp1^+/+^; ^ Dusp1^+/-^*vs. *Dusp1^-/-^* (*,^p<0.05, **,^^p<0.01, ***,^^^p<0.001).

The revised article mentions the auditory phenotype of the heterozygous mouse in the Results section as data not shown, since no differences were found in comparison with wild type mice.

3) The approved name for this gene/protein is now DUSP1 which should be used within the manuscript, rather than MKP1.

In the revised manuscript we use the HGNC nomenclature.

4) Figure 4, if available a lower zoom image of the wholemounts would provide an indication of base to apex gradient in hair cell loss.

The lower zoom images have been taken and will be included in the revised version of the manuscript. Revised Figure 4 has been modified as suggested and microphotographs are presented showing the complete width as they were captured.

5) The RT-qPCR data, although it gives a quantitative assessment of cochlear gene expression, does not take into account the many cell types in the cochlea and an immunofluorescence characterisation of MKP1 and the oxidative stress genes would provide cell specific expression data. Have the authors attempted an immuno-characterisation of MKP1 expression?

The reviewers comment is highly relevant and we should have described better the available data in the literature. MKP1 is a transitory inducible gene (Gass et al., 1996). Thus, we discarded the approach of detecting the protein by immunohistochemistry and this is the reason why expression is shown in the cellular model. The *Mkp1/Dusp1* mRNA expression has been already reported and is available at several Internet repositories; therefore, we decided to reduce the number of mice in the study. Concretely, available data support our results and its expression is low during development (https://www.ensembl.org/Mus_musculus/Gene/ExpressionAtlas?g=ENSMUSG00000024190; r=17:26505590-26508519) in almost all tissues but liver and intestine. RNAseq studies of the organ of Corti available at (gEAR portal of visualization of multi-omic data https://umgear.org/) indicate that it is expressed in the cochlea and vestibule and that expression is enriched in supporting cells. Furthermore, noise has been reported to induce its expression (Alagramam et al., 2014; Kirkegaard et al., 2006). In this connection, we have designed further experiments that will be included in the revised version of the manuscript if we are granted this opportunity.

For the resubmission of the article, we have confirmed the presence of DUSP1 protein in the cochlea by inducing its expresión with noise as already reported (Alagramam et al., 2014; Kirkegaard et al., 2006). Information regarding other studies mentioned above, which indicate its presence in supporting cells in the cochlea and vestibule, has been referenced in the Discussion section.

6) Could they clarify when SYBR green assays were used and when Taqman assays were used by giving Taqman assay IDs. Why were two different internal housekeeping gene assays used?

The SYBR green assays were specified in Table 1 in the manuscript and the IDs of the TaqMan probes will be provided in the revised one. We typically use two housekeeping genes in RTqPCR gene expression experiments and, then, normally *Rplp0* as reference for further calculations. However, in the longitudinal study we observed that the most stable gene expression pattern along time was that of *Hprt1* and therefore it was used in these experiments as reference gene. We can specify this further in the manuscript or, if this possesses a problem, we have the data with both housekeeping genes for all the experiments performed and we can recalculate.

The revised version of the article includes supplementary tables have been prepared specifying the ID of the TaqMan probes and SYBR green assays and are referenced in the Methods section (Supplementary file 2 and Supplementary file 3).

7) It would be interesting to know whether these mice are at increased risk of noise induced hearing loss, although probably not required for publication.

We totally agree with this view, and indeed this was planned. The revised article includes experiments addressing noise exposure of *Dusp1* null mice in a new figure (Figure 7).

8) Although mostly well written there are some typos and grammatical errors particularly in the abstract, which I suspect may have had some last minute changes, which were not proofread. ARHL is quite often mis-written as AHRL throughout manuscript.

Typos have been corrected and the Abstract revised.

9) Discussion section, authors state "oxidative stress as main cause of later problems". Since they have not investigated any other mechanism then this is an over-statement.

The sentence has been rewritten to "oxidative stress is one of the possible causes of later problems” to accommodate the fact that we have not investigated any other mechanism.

Reviewer #3:[…] The capability of MKP1 to regulate oxidative/inflammatory status in the cochlea is innovative and appealing topic. However, the principal limitation of the study is that the authors omitted to describe/speculate on the molecular mechanism that link MKP1 deficiency with the increased oxidative stress/inflammation in the cochlea.

We agree with the reviewers’ concern, although in our opinion, this goes beyond the main message of this manuscript and could be another complete issue, but we agree that is highly recommended to discuss the molecular mechanism involved in the process of hearing loss.

Furthermore, our data encouraged us to explore the molecular mechanism underlying apoptosis induction in cochlea cells. Some speculations could be done based on what it has been reported in other cellular contexts, (Wang et al., 2007; Zhang et al., 2012). First, we hypothesized that inflammatory cytokines could be responsible to generate a retro-alimentation loop. We are addressing this point in vitro, and our data suggest that the absence of MKP1 exacerbate the p38 activation in response to cytokines and, consequently, enhance apoptosis through caspase 3 activation.

Essential revisions:1) The criteria of inclusion of the animals in this study is not clear and this is a crucial point considering that the paper is focused on the age related hearing loss. Namely, it should be explained why the range of age for animal inclusion is wide: 2, 4-5, 8-9, 12 and 16-19 months of age. Considering the lifespan of this mice the differences of 2-3 months are relevant, especially in the old animals. Furthermore, the differences on the phenotype and the auditory function assessment should be illustrated for each group even if major data have been provided in 2-5-8 months of age group.

We understand the reviewers concern and this will be clarified in the reviewed manuscript. We have measured ABR threshold monthly in several cohorts, an example is provided in Author response image 2. Taking into account this parameter, ABR data were grouped afterwards to reduce the number of animals used in further experiments, and also to avoid including redundant information in the manuscript. Certain cohorts were used to obtain specific time points for further analysis. ABR measurements were carried out until knock out mice were profoundly deaf (ABR threshold >90-100 dB SPL). Twelve month-old knock out mice did not show increased mortality rates when compared with wild types, although this was the age-group with the lowest n of those studied here and we will not risk conclusions. A few wild type mice were kept, to know at which age this genotype show profound deafness, therefore we could observe that wild type mice mortality rate increased from 18 months of age on.

**Author response image 2. respfig2:** Monthly evolution of auditory thresholds.

2) Hearing function was assessed from 2 to 12 month-old and MKP1^-/-^ mice show ABR thresholds significantly higher with respect to controls (both for click and tone burst responses in almost all frequencies analyzed) already at 2 months of age; then, hearing function worsened with age compared to MKP1^+/+^ mice. It could be useful to assess hearing thresholds in an earlier time point, in order to demonstrate that MKP1^-/-^ mice show baseline threshold values similarly to controls and that hearing worsen with time compared to MKP1^+/+^ mice. This is relevant issue because the authors state that MKP1 deficiency accelerates the onset and the progression of age-related hearing loss. While, but, based on the current electrophysiological data, it is not clear if MKP1 deficiency also impairs the development of auditory function.

The recommended age to perform ABR is two months; however, we will take earlier time points (ongoing experiments) to confirm if there are differences between genotypes in one-month old mice. The reviewer is correct that a subtle deficiency seems to be there, however all cell types are formed and there are no obvious morphological alterations or cytoarchitecture disorganization. On the other hand, available data suggest a very low expression during development (see above answer to Q5 from reviewer 2).

3) It is not clear how DPOAE was recorded. In the Materials and methods section the authors refer to a previous published procedure (Martinez-Vega et al., 2015) but in this paper DPOAE recording was not performed. Also, OHC count reveals a marked decrease of cell number in the basal turn of MKP1^-/-^ mice, however DPOAE recording was performed by using primary tones for 8-14 kHz. Maybe, it could be useful to analyze also responses for higher frequencies. Also, it is not clear why DP level increased from 4-5 to 8-9 months, when the level of Prestin decreased (as shown in Figure 3B). I would expect a decrease of DPOAE responses in MKP1^-/-^ mice rather than an increase.

The reviewer is right, and the reference was mistaken, the DPOE method was reported in a Martínez-Vega 2015 manuscript entitled “Long-term omega-3 fatty acid supplementation prevents expression changes in cochlear homocysteine metabolism and ameliorates progressive hearing loss in C57BL/6J mice” (Martinez-Vega et al., 2015). The requested DPAOE measurements at higher frequencies will be done. The parameter represented in Figure 3B is DPOAE thresholds, therefore numbers should indeed increase in parallel with the reported loss of OHC. This will be better explained in the revised version of the manuscript to avoid confusions.

No problem. The revised manuscript contains the DPOAE presented as suggested and the correct reference.4) The authors analyzed the level of genes involved in oxidative stress in cochlear samples from both genotypes and the ROS expression (DHE assay) was studied in MEF cells derived from MKP1^-/-^ and MKP1^+/+^ mice. Although the methodology was well conducted, in order to support the major conclusion that MKP1 is essential for the regulation of cochlear redox balance, it could be relevant to establish a direct relationship between cochlear oxidative stress and MKP1 activation in vivo. It could be helpful to perform some quantification (i.e. western blot analyses) on cochlear samples from MKP1^-/-^ and MKP1^+/+^ mice in order to correlate the modifications in cochlear redox status during aging in both genotypes with MKP1 level expression.

Experiments are ongoing to further evaluate cochlear oxidative stress at the protein level.

5) Morphological observations have been performed only in 4-8 months old mice, however, ABRs revealed hearing loss also at 2 months of age in MKP1^-/-^ mice, thus morphological data at earlier time points could be added to support electrophysiological data.

Histological analysis was performed at several ages in both genotypes, although they were not included in the original manuscript. As an example, please see Figure 3—figure supplement 2.

Please, I believe that this criticism is relevant and should be explained.6) Figure 4: The quality of panels a-h could be ameliorated. Maybe fluorescence excitation was saturated, and this does not allow to appreciate the morphological details of the organ of Corti. Also, I suggest modifying DPOAE data presentation. Maybe it could be useful for data interpretation to graph DPOAEs in response to increasing stimulus intensity (showing 2f1-f2 dB SPL in y axis and f1 and f2 input level in the x axis), selecting only one representative frequency.

We agree with these comments, both microphotographs and data have been obtained, and all suggestions will be incorporated in the revised version of the manuscript.

We have improved the quality of a-h panels and added the DPOAE representation as suggested by the reviewer.

[Editors’ note: the author responses to the second round of peer review follow.]

Point 1:In my opinion this criticism remains unsolved in absence of adjunctive experiments. The authors can improve the interpretation of results hypothesizing putative molecular mechanisms leading to activation of oxidative stress/inflammatory pathways, however, providing new evidence on the role of MKP1 molecular mechanism is challenging. The authors suggest a link between MKP1 and p38 activation that could be studied.

Following the reviewers´ suggestion, the revised manuscript includes experiments showing that: (i) DUSP1 absence increases oxidative stress, inflammation and apoptosis in vivo by using a combination of RT-qPCR, western blotting and immunohistochemistry experiments; and (ii) one of the cytokines induced in vivo, TNFα, has been used in cultured MEFs from the null mouse to show that it strongly activates p38 in DUSP1 absence. Furthermore, only in this condition, we observed the activation of caspase 3 that clearly indicated apoptosis program activation. This data has been included in Figure 6—figure supplement 1. All these results taken together reinforce our previous putative models and in addition a putative molecular mechanism leading to activation of oxidative stress is discussed in the revised manuscript.

Point 2:I am not in agreement with this experimental plan, because in my opinion, given the aging processes are one of the major topic of this study, more careful samples of animal age should be done. However, at least I suggest clarifying this point in the manuscript and to report this comment in Discussion section.

A clarification regarding the grouping of animals has been added to the Materials and methods section and data showing the monthly evolution of auditory thresholds presented above have been included in the manuscript under the Results section (Supplementary file 1).

Point 3:I suggest including data regarding ABR recordings at P31. However, considering that the authors state that at 2 months cell types are formed and there are no obvious morphological alterations or cytoarchitecture disorganization, even more so I think that they have to explain the threshold elevation observed in 2 months MKP1 KO animals with respect to Wt age-matched animals.

As suggested by the reviewer, we performed ABR analysis in one-month old mice, no differences were found between genotypes except on the click and 40 kHz stimuli, with an increase of about 8 dB in knock out mice. Data from one-month-old mice has been added to the revised manuscript in a new table (Supplementary file 1).

Point 4I agree, these data have to be added to improve the study relevance.

For the resubmission of the manuscript we have performed noise exposure experiments that link the activation of DUSP1 to this damaging stimulus, which induces oxidative stress and inflammation. Furthermore, we have performed new experiments that strenghten the link between the absence of DUSP1 and the increase in oxidative stress. We have observed by inmunochemistry that null mice show increased levels of nytrotirosine, a product of tyrosine nitration mediated by reactive nitrogen species, at the age of two months (Figure 5—figure supplement 1) and have confirmed by western blotting an increase in the levels of p22phox in 5-month-old null mice. Finally, in the Discussion section, we discuss a putative molecular mechanisms linking DUSP1 to the increase of oxidative stress based on data available in the literature.

Point 5I agree, however, on the basis of these morphological data, as mentioned above, the Authors must to explain why the KO animals show a threshold elevation compared to WT controls at 2 months age. More details could be added.

The reviewer is correct that the difference between genotypes at this young age might be a clue to understand the progressive phenotype. One-month-old mice do not present apparent differences in auditory phenotype, and the 2 month-old histology corroborates the presence of all cell types with no obvious morphological alterations. However, gene expression experiments suggested that a redox unbalance was present at this early age, suggesting that molecular alterations precede the cellular loss phenotype. To asses this hypotheses, we have tested the levels of protein tyrosine nitration caused by increased reactive nitrogen species, indeed the null mouse shows increased 3-NT levels and these data have been included in a new figure of the revised version of the manuscript and further discussed in the text.